# Mapping responses to focal injections of bicuculline in the lateral parafacial region identifies core regions for maximal generation of active expiration

Annette Pisanski[1], Mitchell Prostebby[2], Clayton T Dickson[1,2,3,4,5], Silvia Pagliardini[1,2,4,5]*

[1]Department of Physiology, University of Alberta, Edmonton, Canada; [2]Neuroscience and Mental Health Institute, University of Alberta, Edmonton, Canada; [3]Department of Psychology, University of Alberta, Edmonton, Canada; [4]Department of Anesthesiology and Pain Medicine, University of Alberta, Edmonton, Canada; [5]Women and Children's Health Research Institute, University of Alberta, Edmonton, Canada

*For correspondence:
silviap@ualberta.ca

Competing interest: The authors declare that no competing interests exist.

**Abstract** The lateral parafacial area (pFL) is a crucial region involved in respiratory control, particularly in generating active expiration through an expiratory oscillatory network. Active expiration involves rhythmic abdominal (ABD) muscle contractions during late-expiration, increasing ventilation during elevated respiratory demands. The precise anatomical location of the expiratory oscillator within the ventral medulla's rostro-caudal axis is debated. While some studies point to the caudal tip of the facial nucleus (VIIc) as the oscillator's core, others suggest more rostral areas. Our study employed bicuculline (a γ-aminobutyric acid type A [GABA-A] receptor antagonist) injections at various pFL sites (–0.2 mm to +0.8 mm from VIIc) to investigate the impact of GABAergic disinhibition on respiration. These injections consistently elicited ABD recruitment, but the response strength varied along the rostro-caudal zone. Remarkably, the most robust and enduring changes in tidal volume, minute ventilation, and combined respiratory responses occurred at more rostral pFL locations (+0.6/+0.8 mm from VIIc). Multivariate analysis of the respiratory cycle further differentiated between locations, revealing the core site for active expiration generation with this experimental approach. Our study advances our understanding of neural mechanisms governing active expiration and emphasizes the significance of investigating the rostral pFL region.

## eLife assessment

This manuscript presents experiments that address the question of whether the lateral parafacial area (pFL) is active in controlling active expiration, which is particularly significant in patient populations that rely on active exhalation to maintain breathing (eg, COPD, ALS, muscular dystrophy). This study presents **solid** evidence for a **valuable** finding of pharmacological mapping of the core medullary region that contributes to active expiration and addresses the question of where these regions lie anatomically. Results from these experiments will be of value to those interested in the neural control of breathing and other neuroscientists as a framework for how to perform pharmacological mapping experiments in the future.

## Introduction

The neural control of respiration is a complex physiological process that requires precise coordination among multiple brainstem nuclei. One such nucleus, the preBötzinger complex (preBötC), plays a crucial role in generating the inspiratory rhythm and pattern (*Gray et al., 2001*; *Smith et al., 1991*; *Tan et al., 2008*; *Wang et al., 2014*). In contrast, the lateral parafacial area (pFL), also referred to as pFRG in the initial studies, has emerged as an important brainstem structure containing neurons that are responsible for the generation of active expiration via the recruitment of expiratory musculature, such as the abdominal (ABD) muscles (*de Britto and Moraes, 2017*; *Huckstepp et al., 2015*; *Pagliardini et al., 2011*; *Pisanski and Pagliardini, 2019*). Although the involvement of preBötC in respiration as well as its origin, anatomical location, markers, and anatomical projections has been extensively studied (*Biancardi et al., 2023*; *Del Negro et al., 2018*; *Gray et al., 2001*; *Hayes et al., 2017*; *Tan et al., 2008*; *Wang et al., 2014*; *Yackle et al., 2017*; *Yang and Feldman, 2018*; *Yang et al., 2020*), the expiratory oscillator remains a challenging structure to locate accurately due to the absence of a definite anatomical marker and the lack of its activation during experimental resting conditions (*de Britto and Moraes, 2017*; *Magalhães et al., 2021*; *Pagliardini et al., 2011*). Understanding the precise location and functional properties of the expiratory oscillator is crucial for unraveling the complete neural circuitry underlying respiratory control.

Previous research has employed a variety of tools to study and elucidate this expiratory oscillator that include pharmacological disinhibition and excitation (*Boutin et al., 2017*; *de Britto et al., 2020*; *de Britto and Moraes, 2017*; *Korsak et al., 2018*; *Magalhães et al., 2021*; *Pagliardini et al., 2011*; *Zoccal et al., 2018*), as well as chemogenetic and optogenetic approaches (*Huckstepp et al., 2015*; *Pagliardini et al., 2011*; *Pisanski et al., 2020*). The coordinates used for the latter studies considered the core of the expiratory oscillator proximal to the caudal tip of the facial nucleus (VIIc) (–0.2 mm to +0.5 mm from VIIc). Interestingly, the use of chemogenetics to inhibit pFL at the level of the caudal tip of the facial nucleus (–0.3 mm to +0.3 mm from VIIc) partially inhibited the putative expiratory muscle output, with ABD recruitment still occurring naturally during sleep, albeit at a diminished rate (*Pisanski et al., 2020*). Similarly, chemogenetic inhibition of pFL at +0.5 mm from VIIc decreased the intensity of the ABD signals obtained in response to bicuculline/strychnine injections, but did not silence it entirely (*Huckstepp et al., 2015*). Although both of these elegant chemogenetic studies have extensively contributed to our understanding of the pFL, the existing evidence suggests that the expiratory oscillator may expand beyond the limits of the viral expression achieved in said studies, as proposed by *Huckstepp et al., 2015*. Intriguingly, other research (*Silva et al., 2019*) located its core at more rostral coordinates (+0.3 mm to +1.0 mm from VIIc). This dichotomy in the coordinates used to study the functional properties of the expiratory oscillator demonstrates that the characterization of the anatomical and functional boundaries of pFL are incomplete.

To address this gap, we used focal injections of bicuculline along different rostro-caudal locations of the brainstem as a tool to create a functional map of the pFL. Bicuculline, a competitive antagonist of γ-aminobutyric acid type A (GABA-A) receptors, has been widely employed to study neuronal excitability and functional circuitry in various brain regions, including the pFL (*Pagliardini et al., 2011*). By strategically administering localized volumes of bicuculline at multiple rostro-caudal levels of the ventral brainstem, we aimed to selectively enhance the excitability of neurons driving active expiration, thereby revealing the extension of the pharmacological response and the most efficient site in generating active expiration. Because all the locations studied have been previously shown to produce active expiration (*Pagliardini et al., 2011*; *Boutin et al., 2017*; *de Britto and Moraes, 2017*; *Huckstepp et al., 2015*; *Silva et al., 2019*), we conducted a more holistic analysis of all the respiratory changes induced by bicuculline injections. This analysis included traditional respiratory measures such as tidal volume, minute ventilation, respiratory rate, and oxygen metabolism on top of the ABD changes induced by active expiration. Additionally, using a novel multidimensional cycle-by-cycle analysis specifically developed for this study, we characterized the effect that bicuculline elicited on the different phases of the respiratory cycle at each injection site, as the differences in area under the curve (AUC) of the airflow, ∫DIA EMG, and ∫ABD EMG, as well as the combined differences of these three respiratory signals using a novel phase-plane analysis. This approach enabled us to construct a comprehensive functional map of the pFL in conjunction with anatomical immunostaining techniques.

Our results indicate that the injection of bicuculline produced active expiration at all locations studied (–0.2 mm to +0.8 mm from VIIc). However, the strongest effects in terms of changes in tidal

volume ($V_T$), minute ventilation ($V_E$) and differences in combined respiratory signals were observed at the two most rostral locations (+0.6 mm and +0.8 mm). Similarly, ABD activation lasted longer at these sites and the swiftest onset of the ABD response was observed at the +0.6 mm location. Interestingly, our multivariate analysis of the respiratory cycle permitted further differentiation of the response elicited in the two rostral locations, with the strongest deformations of the respiratory loop during late-expiration (late-E) and post-inspiration (post-I) with injections at the +0.8 mm locations, whereas injections at the level of +0.6 mm producing more pronounced deformations of the respiratory cycle during the inspiratory phase. The use of the multidimensional trajectory map will certainly enhance our understanding of the role that the expiratory oscillator plays in respiratory control. Additionally, it will provide a crucial foundation for future investigations into its modulation, plasticity, and potential therapeutic targets for respiratory disorders.

## Methods

### Experimental subjects

Thirty-five adult male Sprague-Dawley rats weighing 340.4 g±13.2 were used for the study. The rats were housed in a controlled environment with a 12 hr light-dark cycle and were allowed unrestricted access to food and water. All experimental procedures followed the guidelines set by the Canadian Council of Animal Care and received approval from the Animal Care and Use Committee (ACUC) of the University of Alberta (AUP#461).

### Surgical preparation

Prior to the experiment, the rats were initially anesthetized using 5% isoflurane in air for induction, followed by 1–3% isoflurane for maintaining a surgical plane of anesthesia. During this time, we cannulated the femoral vein to facilitate the gradual administration of urethane (1.5–1.7 g/kg body weight) for inducing permanent and irreversible anesthesia. We assessed the depth of anesthesia by monitoring the absence of the withdrawal reflex. We cannulated the trachea and used a flow head connected to a transducer (DP10310N154D, GM Instruments, UK) to detect respiratory flow using a carrier demodulator (CD15, Validyne Engineering, CA, USA). We provided supplemental oxygen (30%) throughout the experiment and connected a gas analyzer (ML 206-1704, ADInstruments, CO, USA) to the tracheal tube. The gas analyzer measured fractional concentration of $O_2$. Based on this and the flow rate at the level of the trachea (minute ventilation), we calculated $O_2$ consumption according to *Depocas and Hart, 1957*. Paired electromyogram (EMG) wire electrodes (AS633, Cooner Wire, CA, USA) were inserted into the oblique ABD and diaphragm (DIA) muscles. The wires were connected to differential amplifiers (model 1700, AM Systems, WA, USA), and we sampled the activity at a rate of 1 kHz using the Powerlab 16/30 system (ML880, ADInstruments, CO, USA). We performed vagotomy by resecting a 2 mm portion of the vagus nerve at the mid-cervical level, and the rats' body temperature was maintained at a constant level of 37±1°C using a servo-controlled heating pad (55-7022, Harvard Apparatus, MA, USA). During the entire experimental procedure, rats breathed spontaneously and end-tidal $CO_2$ was not adjusted through the experimental protocol.

### Bicuculline injections

Following the surgical preparation, we placed the instrumented rats in a prone position on a Kopf stereotaxic frame (model 962, Kopf Instruments, CA, USA). We trimmed and aseptically cleaned the incision area and achieved access to the brainstem by partially removing the occipital bone. We determined the coordinates for injection into the ventral medulla with Bregma positioned 5 mm below lambda. The bicuculline/fluorobeads solution (200 µM in HEPES, bicuculline methochloride, D46M4623V, Sigma-Aldrich, MA, USA; *n*=28) or HEPES/fluorobeads (F8797, Invitrogen, MA, USA) (CTRL, *n*=7) was bilaterally pressure-injected into specific coordinates using a sharp glass micropipette (~30 µm tip diameter) at a rate of 100 nl/min (total volume injected was 200 nl per side). Each rat was injected at 2.8 mm lateral from the midline and at a specific rostro-caudal coordinate based on the following groups: –0.2 mm from the caudal tip of the facial nucleus (VIIc) (*n*=5), +0.1 mm from VIIc (*n*=7), +0.4 mm from VIIc (*n*=5), +0.6 mm from VIIc (*n*=6), +0.8 mm from VIIc (*n*=5), and CTRL (*n*=7). We recorded the physiological responses to the injection for 20–25 min.

## Histology procedures

Upon completion of the experiments, we transcardially perfused the rats with saline (0.9% NaCl), followed by 4% paraformaldehyde in phosphate buffer. We then collected, postfixed, cryoprotected, and sectioned the brains in a cryostat (CM1950, Leica, Germany) to obtain 30 μm thick transverse slices. To prepare the brain tissue for immunohistochemistry, we washed the brain sections multiple times with phosphate-buffered saline (PBS) to remove the cryoprotectant solution (15% sucrose, 15% ethyleneglycol, 0.5% PVP-40 in 0.5X-PBS). All immunostaining procedures were conducted at room temperature. We incubated the sections in a blocking solution (0.3% Triton X-100 and 10% normal donkey serum [NDS] in PBS) for 1 hr to enhance membrane permeability to antibodies and minimize nonspecific binding. After blocking, we incubated the tissue overnight with the primary antibody solution, which consisted of PBS, 1% NDS, and 0.3% Triton X-100. In order to identify specific brain structures relevant to this study, we used the following primary antibodies: anti-choline acetyltransferase (ChAT; goat; 1:800; RRID: AB_20797, EMD Millipore, ON, Canada), anti-PHOX2B (PHOX2B; mouse; 1:100; RRID:AB_2813765, Santa Cruz Biotechnology, TX, USA), anti-cFos (cFos; rabbit; 1:500; RRID:AB_2247211, Cell Signaling Technology, MA, USA), and anti-tyrosine hydroxylase (TH; chicken; 1:800; RRID:AB_570923, EMD Millipore, ON, Canada). The next day, we washed the tissue three times in PBS and incubated it for 2 hr in a secondary antibody solution containing PBS, 1% NDS, and specific secondary antibodies (1:200; Cy3-donkey anti-rabbit, RRID:AB_2307443; Cy5-donkey anti-mouse, RRID:AB_2340820; Cy2-donkey anti-chicken, RRID:AB_2340370; rhodamine-red-donkey anti-goat, RRID:AB_2340423; Jackson ImmunoResearch Laboratories, PA, USA). Following the incubation with the secondary antibodies, we washed the sections three times and mounted them with Fluorsave mounting medium (345789-20 ML, EMD Millipore, ON, Canada). We observed the slides under an Evos FL fluorescent microscope (AMF4300, Thermo Fisher Scientific, MA, USA) and acquired TIFF files for cell counting and analysis of the rostro-caudal, mediolateral, and dorsoventral coordinates of the injection sites using ImageJ. We examined serial sections with a 120 μm interval, spanning from 400 μm caudal to 1000 μm rostral to the caudal tip of the facial nucleus (VIIc). To group the animals according to the location of the injections, we calculated a 3D distance from the VIIc using the rostro-caudal, mediolateral, and dorsoventral coordinates of each injection site.

## Data acquisition and analysis

EMG signals were amplified at ×10,000 and filtered between 300 Hz and 1 kHz (model 1700, AM Systems, WA, USA). The data was sampled at a rate of 1 kHz (using an automatic anti-aliasing filter at 500 Hz) using the PowerLab 16/35 acquisition system and analyzed using LabChart7 Pro (ADInstruments, CO, USA), Excel 2013, Origin 9 (OriginLab Corp., Northampton, MA, USA), as well as custom scripts written in MATLAB (The Mathworks Inc). All raw EMG data were digitally rectified and integrated using a time-constant decay of 0.08 s. Respiratory airflow was further used to determine respiratory rate, $V_T$, and $V_E$. The tidal volume was obtained by integrating the airflow amplitude and converting it to milliliters using a five-point calibration curve (0.5–5 ml range). The coupling between ABD and DIA signals was measured as a ratio and analyzed by quantifying the number of bursts of activity observed for the ABD and DIA EMG signals during the first 10 min of the response, excluding time bins at end of the response (due to fading and waning of the ABD activity in those instances). The ABD delay was calculated as the time between the end of the second injection and the beginning of the ABD response (negative values mean that the response started following the first injection before the second injection occurred). Whereas ABD duration was calculated by measuring the time between the first and the last ABD burst elicited by bicuculline.

Airflow and integrated EMG signals (*Figure 1A*) were standardized across rats by first zeroing signals in the rest period phases between breaths as recorded during baseline conditions and then converting the remaining amplitude values into standard deviation units (SD) based on a time-collapsed amplitude distribution. Baseline data was averaged over a period of 5 min. Experimental data was tracked in time bins of 2 min duration from the baseline period prior to injections and spanned 20 min of recording post-injection. Mean-cycle measurements for each signal were computed by averaging across all cycles within a given time bin (~300 cycles in baseline, ~100 cycles per response time bin). We then used the average calculations of respiratory rate (RR), tidal volume ($V_T$), minute ventilation ($V_E$), expiratory ABD amplitude, expiratory ABD area, $VO_2$, $V_E/VO_2$ to obtain values relative to the baseline period. Peak responses were identified as the time bin that produced the strongest changes

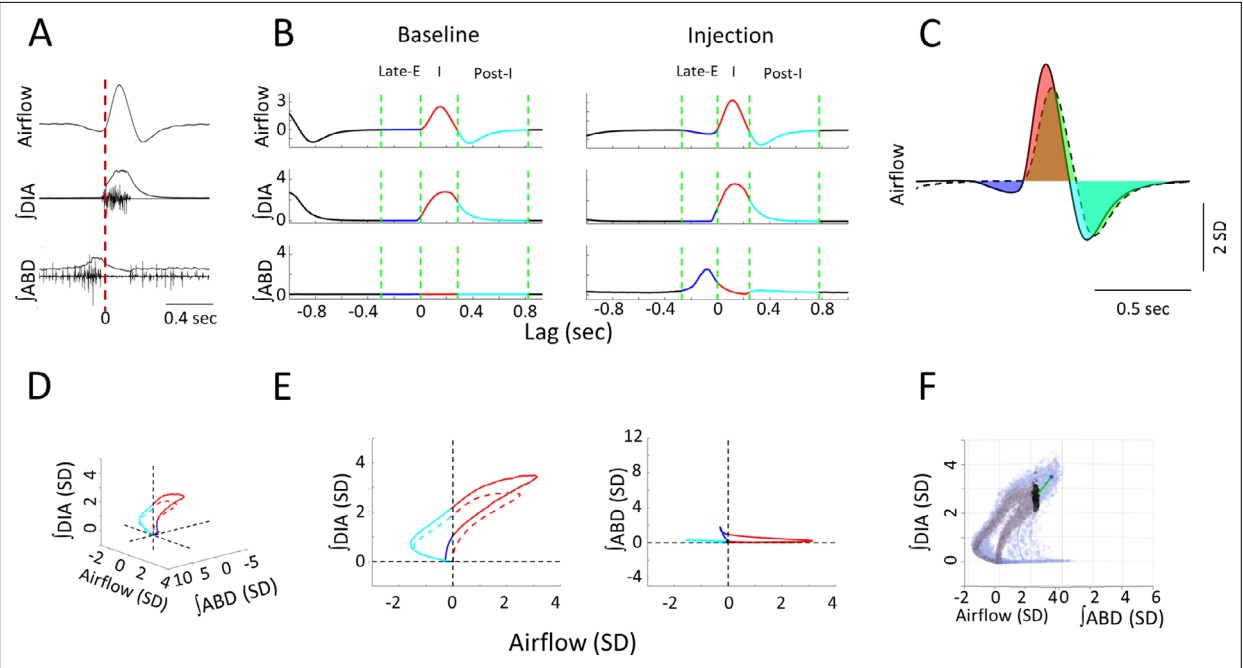

**Figure 1.** Measures used in quantifying respiratory responses to bicuculline injections. (**A**) The time course of raw, and integrated electromyogram (EMG) signals relative to recorded airflow during a sample breath. Vertical red line indicates the onset of inspiration as defined by positive airflow. (**B**) Respiratory phases within a mean-cycle computed during baseline and post-injection for each recorded signal. Green lines mark the boundaries of each phase of the respiratory cycle, including the late-expiratory (late-E) (blue), inspiratory (red), and post-inspiratory (post-I) (cyan) phases. (**C**) Sample calculation of normalized area under the curve. Area under the baseline mean-cycle (dashed line, area in green) in each phase is subtracted from the corresponding colored area under the response mean-cycle (solid line, late-E: blue, inspiration: red, post-I: cyan) to give a measure of how inspiratory airflow has changed relative to baseline. (**D**) 3D representation of a representative baseline (dashed line) and response (solid line) mean-cycle. Each timepoint is a dot in 3D space defined by its airflow, ∫DIA EMG, and ∫ABD EMG measurements. Colors indicate respiratory sub-periods as in B. Dashed lines indicate the origin for each measure. (**E**) 2D projection of the mean-cycle in D. As per the colors in B, points on the left-hand side indicate expiration, points on the right-hand side indicate inspiration. (**F**) Sample calculation of Euclidean and Mahalanobis distance measures for representative mean-cycles during the baseline (gray) and response (orange). PinkGreen line indicates the Euclidean distance between the response and baseline at a single timepoint. Black indicates the distribution of baseline values at the same timepoint used to calculate the Mahalanobis distance. Mean-cycles have been rotated relative to D to expose these distances to the viewer. ABD, abdominal; DIA, diaphragm.

The online version of this article includes the following source data for figure 1:

**Source data 1.** Measures used in quantifying respiratory responses to bicuculline injections, dataset for *Figure 1A–E*.

**Source data 2.** Measures used in quantifying respiratory responses to bicuculline injections, dataset for *Figure 1F* (abdominal electromyogram [EMG] dataset).

**Source data 3.** Measures used in quantifying respiratory responses to bicuculline injections, dataset for *Figure 1F* (respiratory flow dataset).

**Source data 4.** Measures used in quantifying respiratory responses to bicuculline injections, dataset for *Figure 1F* (diaphragm electromyogram [EMG] dataset).

relative to baseline. To assess the changes within each of the phases of the breathing cycle, each mean-cycle was subdivided into the late-E, inspiratory, post-I, and resting phases using the zero-crossings of the mean airflow signal (green lines, *Figure 1B*). The inspiratory phase begins with an inward (positive going) intake of air as airflow increases above zero and ends when airflow crosses zero toward the negative direction. This negative going deflection begins the post-I phase, which lasts until airflow returns to zero during the rest period. The late-E phase is defined in two ways. In cases where a significant negative deflection in airflow was observed previous to inspiration, the late-E phase was determined to have begun when airflow decreased below zero and ended when airflow crossed back above zero to begin the subsequent inspiratory phase. When airflow did not decrease below zero prior to inspiration, as was often the case during baseline conditions, the late-E phase was determined as the last quarter of the expiratory period (*Figure 1B*). The AUC was measured during baseline and was subtracted from the corresponding AUC of the response for each time bin (*Figure 1C*). This AUC measure was computed as the sum of the signal in a given respiratory phase as all signals were

sampled at the same rate. Note that areas calculated below the zero (0) line, as would be expected from a negative airflow during expiration, yields negative AUC values. Any changes to the duration of the three respiratory phases, as well as changes to the entire respiratory period, were independently assessed using the triggers described above.

We further contrasted the overall changes to the three respiratory signals elicited across injection locations by combining them into a multidimensional (3D) space (**Figure 1D**). In a manner similar to respiratory pressure-volume loops, each point in time is defined as a dot in this 3D space based on the three simultaneous physiological measurements of respiration we recorded (airflow, together with the ∫DIA EMG, and ∫ABD EMG). Each respiratory cycle could then be plotted as a trajectory of points in this space, represented as a loop. Example loops for the baseline and the first 2 min time bin of the elicited response are shown in **Figure 1D** and are also projected onto separate 2D planes in **Figure 1E**. Our approach was to measure how the trajectories of response loops changed relative to the baseline loop during each phase of the respiratory cycle, as per the color scheme (same as in **Figure 1B**). Given these loops may be deformed as a result of changes to any one of the three signals which form the basis of this 3D space, this method determines the *total* extent to which breathing is altered in response to stimulation. We quantified changes to respiratory trajectories elicited by bicuculline injection using the sum of the 'straight-line' or 'Euclidean' distances between points on the response and baseline loops in each respiratory phase. An example of the Euclidean distance between the peak of inspiration in baseline and response loops is shown in **Figure 1F** (pink line). To compute this distance for all timepoints, we resampled the response to ensure that that baseline and response mean-cycles had the same number of comparison phase points in a given respiratory phase. Distances were then calculated between these baseline and response loops at each timepoint $t$ using the Euclidean distance formula:

$$Euclidean\ Distance_t = \sqrt{AirDiff_t^2 + DiaDiff_t^2 + AbdDiff_t^2}$$

where $AirDiff_t$, $DiaDiff_t$, and $AbdDiff_t$ are the differences between the response and baseline mean-cycles at timepoint $t$. Averaging the distances measured at every timepoint in a given respiratory phase yields the mean Euclidean distance measure .

We extended our multivariate analysis of respiration to account for the cycle-by-cycle variability in the recorded signals. At each timepoint, we compared how far away the response mean-cycle is from the baseline mean-cycle relative to the covariance across all baseline cycles. Thus, this measure can be interpreted as having units of standard deviation, as it describes how likely it is to observe the response given the distribution of baseline measurements gathered across many respiratory cycles. This measure is known as the Mahalanobis distance and is exemplified as the pink distance relative to the black cloud of points in the baseline mean-cycle of **Figure 1F**. Similarly to above, this distance is computed for each timepoint $t$:

$$Mahalanobis\ Distance_t = \sqrt{(R_t - \mu_t) \sum_t^{-1} (R_t - \mu)'}$$

where $R_t$ is the response mean-cycle at timepoint $t$, $\mu_t$ is the baseline mean-cycle at timepoint $t$, and $\Sigma_t$ is the covariance of all baseline cycles at timepoint $t$. Similar to the Euclidean distance above, we the calculated Mahalanobis distances across all timepoints with each respiratory phase and then averaged them across cycles to assess the mean Mahalanobis distance.

## Statistics

We tested the assumptions of homogeneity of variance and normal distribution using the Brown-Forsythe ($\alpha=0.05$) and Shapiro-Wilk ($\alpha=0.05$) tests, respectively. Comparisons between injection locations were carried out across time bins using a two-way repeated measures analysis of variance (two-way RM ANOVA; one repeated factor; $\alpha=0.05$) where appropriate. In case of significant main effects, we conducted pairwise multiple comparisons post hoc procedures (Bonferroni method; $\alpha=0.05$). Analysis of the peak response across injection locations was done using an analysis of variance (one-way ANOVA; $\alpha=0.05$) in the case of absolute values, or a one-way RM ANOVA ($\alpha=0.05$) in the case of values reported relative to baseline conditions. Upon finding significant main effects, we

used Tukey or Bonferroni post hoc test ($\alpha$=0.05) respectively. Effect size was computed within the analyses of variance using $\eta^2$ (eta squared). In cases where assumptions of normality were rejected (p<0.05), we compared responses across injection location via a Kruskal-Wallis test and following with a post hoc Dunn's test, adjusting the alpha value for multiple comparisons with Sidak's method.

## Results

### Histological analysis

For each rat, we precisely identified the core of the injection sites by assessing the rostro-caudal coordinates obtained from sections containing fluorobeads. Based on the core coordinates, experiments were divided into five groups in which injection spanned from –0.2 mm caudal to the tip of the facial nucleus (VIIc) to +0.8 mm rostral to VIIc (as illustrated in *Figure 2B*). All injections were consistently positioned ventrolateral to the facial nucleus, and there was no overlap observed between the fluorobead-marked injection sites and PHOX2B+ cells within the more ventro-medial retrotrapezoid nucleus (RTN), as depicted in *Figure 2A and B*. Additionally, the injection sites were confirmed not to overlap with the soma of TH+ cells in either the caudal C1 or the more rostral A5 areas, as shown in *Figure 2A and B*.

To assess the cellular responses surrounding the injection sites, we quantified both cFos+ and cFos+/PHOX2B+ cells within a defined region surrounding the fluorobead-marked areas in the pFL (as outlined in *Figure 2A*). In rats injected with HEPES buffer (control group), we observed an average activation of 44.7±4.0 cells per hemisection along the rostro-caudal axis of the ventral medulla (*n*=7). Among these activated cells, an average of 2.3±1.0 cells per hemisection were also positive for PHOX2B, as illustrated in the inset of *Figure 2C*. In contrast, rats injected with bicuculline displayed a distinct response pattern, characterized by a peak activation with an average of 104.5±5.1 cFos+ cells localized at the core of each injection site (–0.2 mm=89.7; +0.1 mm=123.1; +0.4 mm=110.2; +0.6 mm=98.3; +0.8 mm=101.2; *Figure 2C*). Importantly, the number of cFos+ cells decreased to 44.8±1.2 per section beyond the boundaries of the injection area for each experimental group. On the other hand, the number of cFos+/PHOX2B+ cells remained consistent between the control (CTRL) animals (2.3±1.0 cFos+/PHOX2B + cells per hemisection) and those injected with bicuculline (2.4±0.4 cFos+/PHOX2B+ cells). These findings strongly suggest that bicuculline specifically activated disinhibited cells within the vicinity of the injection sites which spread ~300 μm (*Figure 2C*, horizontal lines) and did not activate PHOX2B+ cells in the RTN area, beyond their baseline level of activity.

### Characteristics of ABD signals across the rostro-caudal axis of pFL

We successfully elicited ABD responses with the administration of bicuculline at all tested injection sites. The ABD response was most reliably triggered at locations rostral to the tip of VIIc. This observation is supported by the fact that the caudal location (–0.2 mm from VIIc) elicited responses in only three out of five rats, whereas all other groups consistently induced responses in all rats (*n*=5–7 rats/ group). The elicited response was characterized by a late-E component in the ∫ABD EMG trace across all experimental groups (*Figure 2D*), as well as a downward deflection in airflow, signifying forced exhalation of the respiratory reserve volume preceding inspiration (i.e. active expiration; *Figure 2D*). Remarkably, in the most rostral groups (+0.6 mm and+0.8 mm rostral to VIIc), the ∫ABD EMG traces exhibited a tonic expiratory component, which preceded the late-E peak and was absent in the most caudal locations (–0.2 mm, +0.1 mm, and +0.4 mm). Additionally, five out of six rats in the group with injection sites at +0.6 mm rostral to VIIc exhibited a post-I ABD peak, a feature that was absent in all other groups (*Figure 2D*).

### Temporal dynamics of the ABD response reveal a longer-lasting and quicker response at rostral locations

The temporal characteristics of the ∫ABD EMG signal elicited by bicuculline injections at different rostro-caudal locations exhibited notable variations. Overall, responses had a shorter duration in the most caudal locations (–0.2 mm; +0.1 mm; +0.4 mm) compared to the rostral locations (+0.6 mm; +0.8 mm; *Figure 3A*). In all groups, the ABD response initiated following the second bicuculline injection, except at location +0.6 mm, where it commenced following the first injection and became more pronounced with the second injection (*Figure 3A*). Across all groups, the most robust ABD responses,

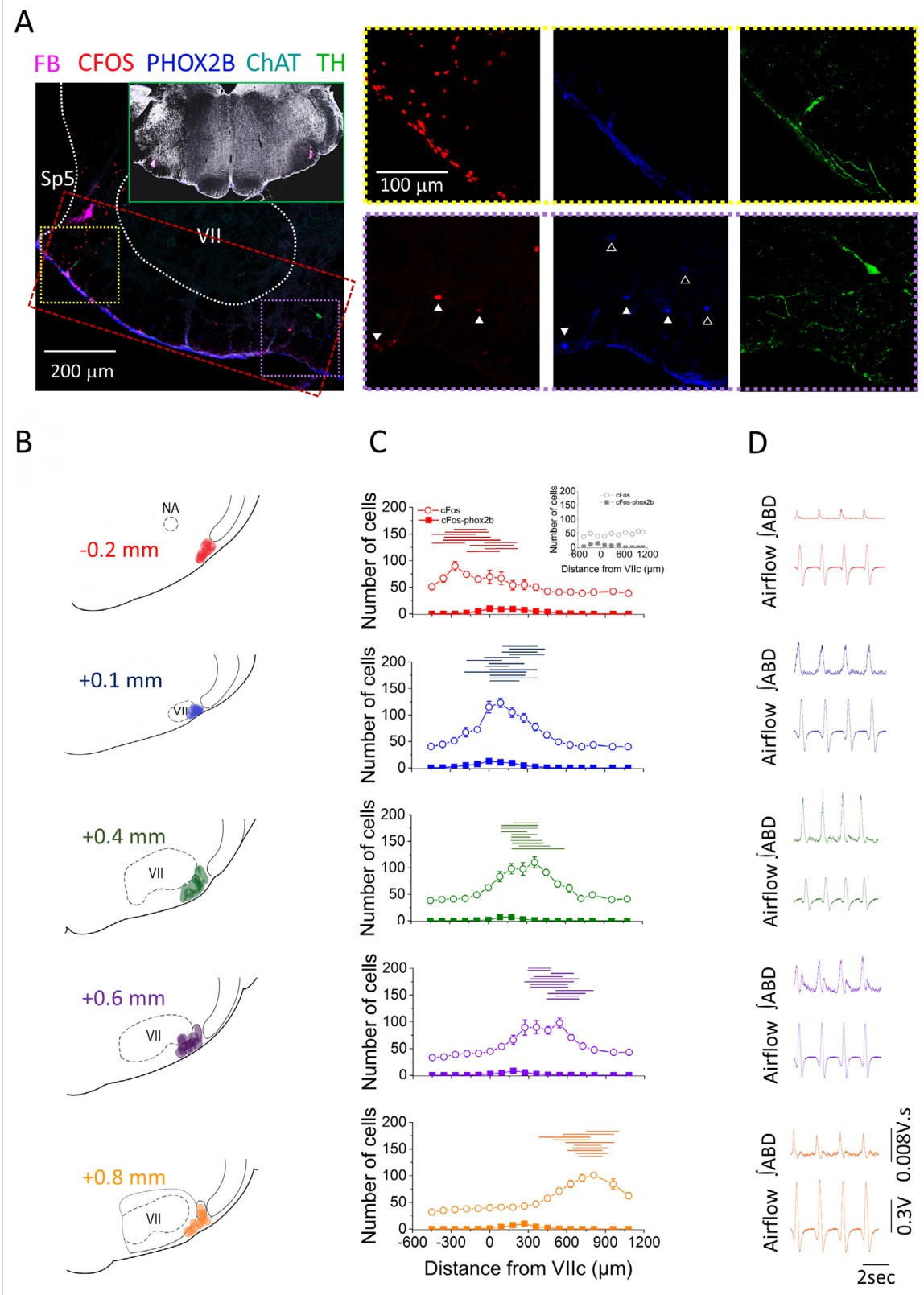

**Figure 2.** Location of bicuculline injections along the rostro-caudal axis of the ventral medulla. (**A**) Representative confocal microscope images of immunohistochemistry performed at the injection sites. Red: cFos, cyan: ChAT, blue: PHOX2B, green: TH, magenta: fluorobeads. Inset shows bilateral injection sites at low magnification. Red rectangle is a representative example of the area used for cell counting. Yellow and purple squares are the areas represented on the right panel of A, notice the abundance of CFOS+ cells on the lateral area near the injection site (yellow square) compared

*Figure 2 continued on next page*

*Figure 2 continued*

to the fewer CFOS+ cells in the medial area (purple square). Closed white triangles are pointing out examples of CFOS-PHOXB+ cells, whereas open white triangles are pointing out examples of PHOX2B+ cells that were CFOS-negative. Magnification is the same for all images in the right panels of A. (**B**) Schematic representation of the core of the injection sites and the group attribution, according to proximity of injection sites among experimental rats (–0.2 mm, *n*=5; +0.1 mm, *n*=7; +0.4 mm, *n*=5; +0.6 mm, *n*=6; +0.8 mm, *n*=5). (**C**) Cell counting color coded according to the groups defined in B (CTRL, *n*=7; –0.2 mm, *n*=5; +0.1 mm, *n*=7; +0.4 mm, *n*=5; +0.6 mm, *n*=6; +0.8 mm, *n*=5). Data points represent the mean ± SEM of cells counted per hemisection at rostro-caudal locations ranging from –0.5 mm to +1.0 mm from VIIc. Open circles: CFOS+ cells; closed squares: CFOS/PHOX2B+ cells. Inset in group –0.2 mm is the cell count obtained for the CTRL group. Horizontal lines above the cell count graphs represent the rostro-caudal extension in which fluorobeads were observed for each injection site. (**D**) Representative examples of ∫ABD EMG and raw airflow traces obtained after injection of bicuculline (color coded based on the groups defined in B). ABD, abdominal; EMG, electromyogram.

The online version of this article includes the following source data for figure 2:

**Source data 1.** Location of bicuculline injections along the rostro-caudal axis of the ventral medulla.

as measured by ∫ABD EMG amplitude, were observed within the first 2 min of the response, declining to zero in the last 2 min (18 min post-injection) in the most caudal groups (–0.2 mm; +0.1 mm; +0.4 mm). In contrast, these responses remained present, albeit weaker, in the most rostral groups (+0.6 mm; +0.8 mm) after 20 min from injection (*Figure 3B and C*).

The coupling of the ∫ABD EMG and ∫DIA EMG peak signals, which reflects the robustness of the response and the coupling between inspiration and active expiration, was notably weaker at the most caudal location (–0.2 mm=0.6±0.2) in comparison to the most rostral groups (+0.4 mm=0.96±0.02; +0.6 mm=0.89±0.05; +0.8 mm=0.97±0.02; one-way ANOVA, p=0.015; $\eta^2$=0.43; Tukey –0.2 mm vs +0.4 mm: p=0.024; –0.2 mm vs +0.6 mm: p=0.048; –0.2 mm vs +0.8 mm: p=0.029; *Figure 3D*). Similarly, the ABD response duration was longer at the two most rostral locations (+0.6 mm=17.6 ± 2.7 min; +0.8 = 17.1±3.3 min) compared to the most caudal group (–0.2 mm=2.4 ± 1.1 min; one-way ANOVA, p=0.043; $\eta^2$=0.41; Tukey –0.2 mm vs +0.6 mm: p=0.048; –0.2 mm vs +0.8 mm: p=0.041; *Figure 3E*). Lastly, the group with injection sites located at +0.6 mm initiated responses sooner than all other groups (–0.2 mm=20.3±13.4 s; +0.1 mm=32.5±20.6 s; +0.4 = 40.1±28.7 s; +0.8 = 23.1±19.8 s), with an average response delay of 88.7±32.3 s previous to the second injection (but after the first one), which is represented as negative values (one-way ANOVA, p=0.041; $\eta^2$=0.57; Tukey –0.2 mm vs +0.6 mm: p=0.039; +0.1 mm vs +0.6 mm: p=0.040; +0.4 mm vs +0.6 mm: p=0.045; +0.8 mm vs +0.6 mm: p=0.049; *Figure 3F*). In summary, these results underscore that injections at the most rostral locations (+0.6 mm and +0.8 mm) elicit more robust and prolonged ABD responses than the caudal location (–0.2 mm). Furthermore, bicuculline injections at +0.6 mm from VIIc initiated the fastest response, suggesting proximity to the cells responsible for ABD recruitment following bicuculline injection.

## Bicuculline injection elicited stronger respiratory effects at the rostral locations

The ABD response elicited by bicuculline injection generated a late-E downward inflection in airflow that was absent in baseline conditions (*Figure 4A*). We measured the late-E peak amplitude and area to assess the strength of the response across different rostro-caudal locations. On average, the late-E peak amplitude and area values reached a maximum between 2 min and 4 min post-second injection for all rostro-caudal locations except the most rostral injection site (+0.8 mm), which peaked between 6 min and 8 min post-second injection (*Figure 4B and D*). Although the maximum late-E peak amplitude in the two most rostral groups (+0.6 mm=–0.033±0.007 V; +0.8 mm=–0.027±0.011 V) were not different from the remaining locations (–0.2 mm = –0.014±0.004 V; +0.1 mm=–0.021±0.007 V; +0.4 mm=–0.023±0.004 V; one-way ANOVA, p=0.41; *Figure 4C*), the large effect size ($\eta^2$=0.18) would suggest that further investigation is warranted. Similarly, the late-E peak area in the two most rostral locations (+0.6 mm=–0.0056±0.0010 V·s; +0.8 mm=–0.0047±0.0019 V·s) were not different from the rest of the injection sites (–0.2 mm = –0.0014±0.0007 V·s ; +0.1 mm=–0.0035±0.0015 V·s; +0.4 mm=–0.0038±0.0011 V·s; one-way ANOVA, p=0.30; *Figure 4E*), but there was a large effect size ($\eta^2$=0.2).

The injection of bicuculline along the rostro-caudal axis of the ventral medulla induced a drop in respiratory frequency in all the injection sites tested (*Figure 4F*), but this drop in frequency was only significant in comparison to baseline in the two most caudal groups (one-way RM ANOVA, p=0.003; $\eta^2$=0.46; Bonferroni: –0.02 mm baseline vs injection, p=0.03; +0.1 mm baseline vs injection, p=0.005;

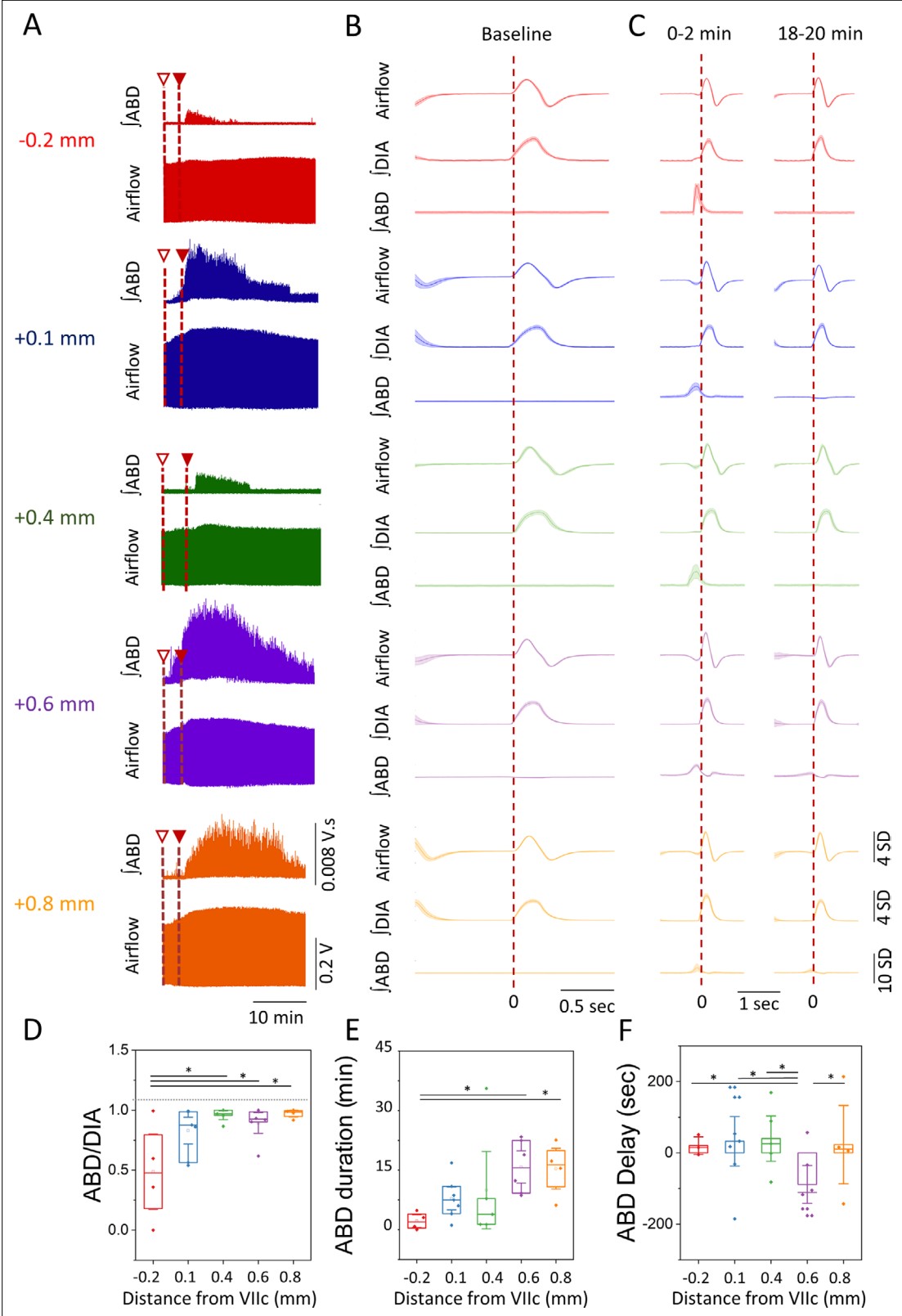

**Figure 3.** Temporal characteristics of the ABD response elicited after bicuculline injection. (**A**) Representative examples of raw traces of the airflow and ∫ABD EMG signals for the entire duration of the response obtained at each injection site following the first (open triangle) and second injection (closed triangle) of bicuculline (examples are color coded based on the groups obtained in *Figure 2B*). (**B–C**) Representative mean-cycles for the airflow, ∫DIA EMG, and ∫ABD EMG during baseline (**B**) and during the first 2 min and the last 2 min of the response (**C**) (traces are color coded based on the groups

*Figure 3 continued on next page*

*Figure 3 continued*

defined in *Figure 2B*). Shaded areas indicate standard deviation of each signal. (**D**) Coupling of the ∫ABD EMG and ∫DIA EMG following the injection of bicuculline at different rostro-caudal locations. (**E**) Duration of the ABD response elicited. (**F**) Delay in the onset of the ABD response following the second injection of bicuculline. Sample size for plots on D–F are as follows: –0.2 mm, *n*=5; +0.1 mm, *n*=7; +0.4 mm, *n*=5; +0.6 mm, *n*=6; +0.8 mm, *n*=5. Boxplots represent the median, interquartile range, as well as the minimum and maximum values. Significance levels were obtained through a one-way analysis of variance (ANOVA) followed by a Tukey test, p<0.005. ABD, abdominal; DIA, diaphragm; EMG, electromyogram.

The online version of this article includes the following source data for figure 3:

**Source data 1.** Temporal characteristics of the abdominal (ABD) response elicited after bicuculline injection (data points for traces presented in *Figure 3B and C*).

**Source data 2.** Temporal characteristics of the abdominal (ABD) response elicited after bicuculline injection (*Figure 3D–F* dataset).

*Figure 4G*). The respiratory rate reached a minimum value of 37.08±1.36 bpm (12% drop from baseline) at 2 min post-injection in the most caudal location (–0.2 mm), at 4 min for the middle groups (+0.1 mm=37.2±2.0 bpm – 17% drop from baseline; +0.4 mm=40.1±1.6 bpm – 9% drop from baseline) and at 6 min for the most rostral injection sites (+0.6 mm=38.6±2.6 bpm – 10% drop from baseline; +0.8 mm=39.1±2.2 bpm – 11% drop from baseline; *Figure 4F*). Interestingly, the duration of inspiration during the response decreased in all groups relative to baseline respiration ($T_i$ response = 0.279±0.034 s, $T_i$ baseline = 0.318±0.043 s, Wilcoxon rank sum: *Z*=3.24, p=0.001). Contrary to this decrease in inspiratory duration, the total expiratory time increased in all groups and remained elevated compared to baseline ($T_E$ response = 1.313±0.188 s, $T_E$ baseline = 1.029±0.161 s, Wilcoxon rank sum: *Z*=4.49, p=0.001). Overall, these results suggest that the most caudal locations (–0.2 mm and +0.1 mm) experienced a faster and more drastic drop in respiratory frequency following bicuculline injection, and this reduction in frequency was driven by an increase in expiratory period accompanied by a reduction in the inspiratory time.

The amount of inspired air per each breath, measured as $V_T$, increased when compared to baseline in all the injection sites tested except the most caudal location (–0.2 mm from VIIc) (one-way RM ANOVA, p=0.01; $\eta^2$=0.44, *Figure 4H and I*). In all the groups, the increase in $V_T$ peaked at 4 min post-injection (*Figure 4H*). Of particular interest is that the peak $V_T$ observed in the +0.6 mm location ($V_T$ = 10.8±0.5 ml/kg – 29% increase from baseline) was higher than that observed in the two most caudal groups (–0.2 mm=7.0±0.4 ml/kg – 8% increase from baseline and +0.1 mm=8.0±0.3 ml/kg – 16% increase from baseline; one-way ANOVA, p=0.01; Bonferroni, p=0.03; *Figure 4I*). On the other hand, $V_E$ decreased compared to baseline in the most caudal group (–0.2 mm = 255.4±16.2 ml × min$^{-1}$ × kg$^{-1}$ – 12% drop from baseline, one-way RM ANOVA, p=0.001; $\eta^2$=0.59), whereas at the +0.6 mm location, $V_E$ increased following bicuculline injection ($V_E$ = 414.3±22.7–16% increase from baseline, one-way RM ANOVA, p=0.001; $\eta^2$=0.59, *Figure 4J and K*). Interestingly, the change produced in $V_E$ after injection of bicuculline at the most rostral locations (+0.4 mm; +0.6 mm and +0.8 mm) was larger than that produced after injection at –0.2 mm from VIIc (one-way ANOVA, p=0.001, $\eta^2$=0.59; Bonferroni, p=0.0004; *Figure 4K*). These results suggest that the $V_E$ was mostly driven by the drop in respiratory frequency at the two most caudal locations (–0.2 mm and +0.1 mm), whereas $V_T$ drove the increase in $V_E$ observed in the most rostral groups (+0.4 mm; +0.6 mm; +0.8 mm).

## Injection of bicuculline decreased oxygen consumption at the rostral locations

To evaluate the metabolic effects of the respiratory changes induced by bicuculline injections, we measured the amount of oxygen consumption ($V_{O2}$). Overall, $V_{O2}$ remained similar to baseline at the three most caudal locations (–0.2 mm; +0.1 mm; and +0.4 mm; *Figure 4L and M*) and dropped from baseline only at the most rostral sites (+0.6 mm and +0.8 mm) (one-way RM ANOVA, p=0.0001; $\eta^2$=0.59). The minimum $V_{O2}$ achieved at the rostral groups (+0.6 mm=11.8±0.8 ml × min$^{-1}$ × kg$^{-1}$ – 33% drop from baseline and +0.8 mm=12.2±6.1 ml × min$^{-1}$ × kg$^{-1}$ – 25% drop from baseline) was also different from the values achieved at the most caudal locations (–0.2 mm = 17.3±3.3 ml × min$^{-1}$ × kg$^{-1}$ – 10% drop from baseline; +0.1 mm=17.6±1.3 ml × min$^{-1}$ × kg$^{-1}$ – 17% drop from baseline; one-way RM ANOVA, p=0.0001, $\eta^2$=0.59, Bonferroni, p=0.005; *Figure 4M*). Interestingly, the ratio of $V_E/V_{O2}$ increased only at the two most rostral locations (one-way RM ANOVA, p=0.001; $\eta^2$=0.55; *Figure 4N–O*), peaking at 6–8 min post-second injection (+0.6 mm=34.8±3.0–73% increase from BL;

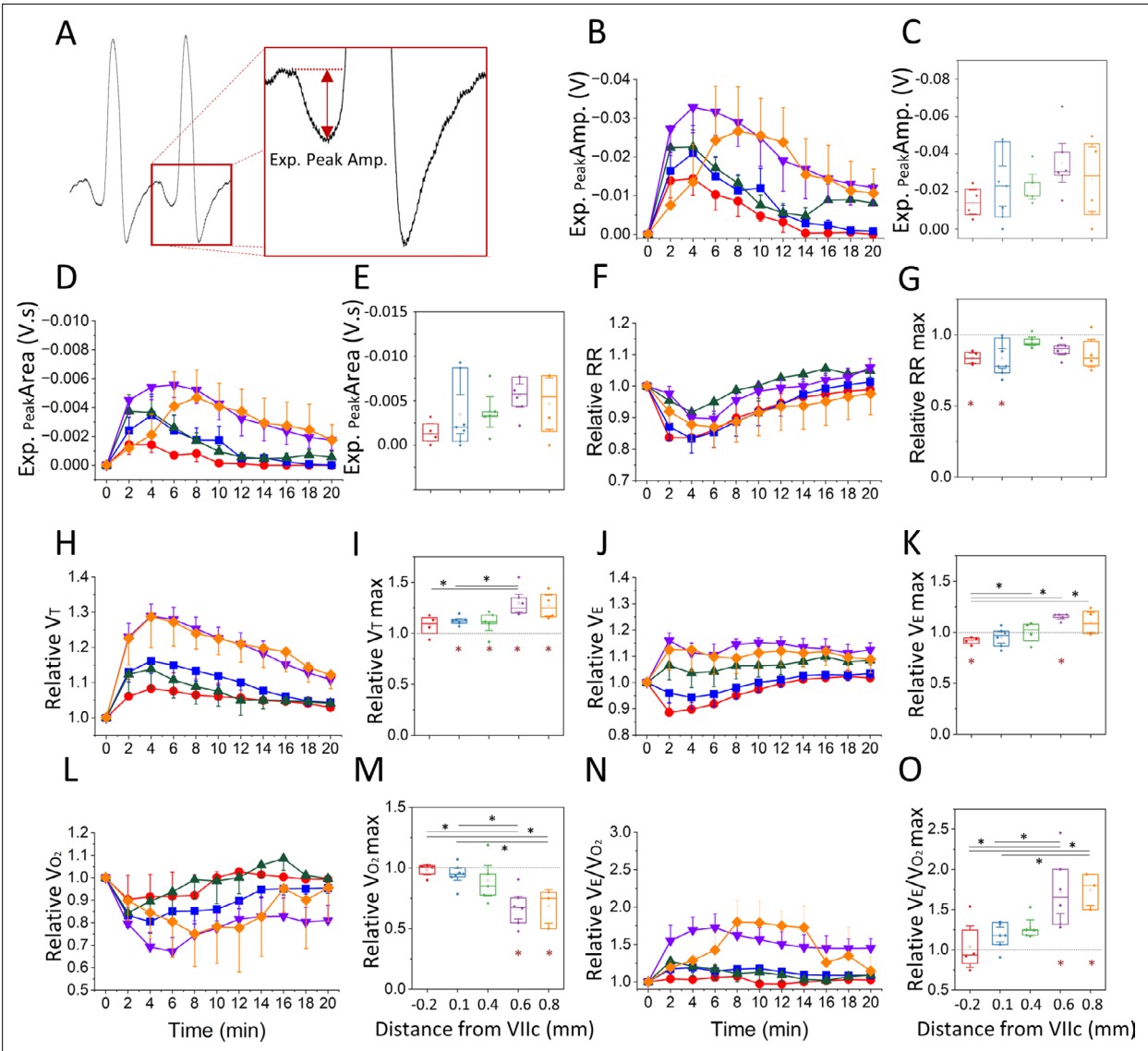

**Figure 4.** Respiratory changes elicited following bicuculline injection. (**A**) Airflow trace depicting the late-expiratory (late-E) inward airflow inflection induced during abdominal (ABD) recruitment, as well as the measure of expiratory peak amplitude used in **B**. (**B, D**) Expiratory peak amplitude (**B**) and area (**D**), throughout the duration of the post-injection period for each group (color coded based on *Figure 2B*). Values represent the mean ± SEM at each time bin. (**C, E**) Maximum expiratory peak amplitude (**C**) and area (**E**) obtained at each injection site. (**F, H, J, L, N**) Relative change in respiratory rate (**F**), tidal volume (**H**), minute ventilation (**J**), oxygen consumption (**L**), and $V_E/V_{O_2}$ (**N**) with respect to baseline throughout the duration of the post-injection period for each group (color coded based on *Figure 2B*). Values represent the mean ± SEM at each time bin. (**G, I, K, M, O**) Maximum relative change in respiratory rate (**G**), tidal volume (**I**), minute ventilation (**K**), oxygen consumption (**M**), and $V_E/V_{O_2}$ (**O**) observed at each injection site. Sample size for plots on B–O are as follows: –0.2 mm, *n*=5; +0.1 mm, *n*=7; +0.4 mm, *n*=5; +0.6 mm, *n*=6; +0.8 mm, *n*=5. Boxplots represent the median, interquartile range, as well as the minimum and maximum values. Significance levels were obtained through a one-way repeated measures analysis of variance (ANOVA) followed by a Bonferroni test, p<0.005 (black asterisks represent significant comparisons between injection sites, red asterisks represent significant comparisons relative to baseline).

The online version of this article includes the following source data for figure 4:

**Source data 1.** Respiratory changes elicited following bicuculline injection (*Figure 4* dataset).

+0.8 mm=38.7±12.8–82% increase from baseline; *Figure 4N*). In summary, these results suggest that bicuculline injection and the resulting respiratory effects led to a reduction in oxygen consumption in the most rostral locations. Furthermore, the reduction in $V_{O_2}$ paired with the increases in $V_E$ observed at the most rostral injection sites, resulted in hyperventilation, an effect which was not observed in the two most caudal groups.

## Rostral bicuculline injection induces more prominent changes to all phases of the respiratory cycle

To further differentiate the responses elicited by bicuculline injections, we subdivided our analysis by respiratory phase (late-E, inspiratory, and post-I) and measured the differences in the area of respiratory signals (airflow, ∫DIA EMG, and ∫ABD EMG) as per the algorithm described in the Methods section (*Figure 5A–C*).

Injections at all locations induced negative deflections in the airflow signal (*Figure 5D*) and were concomitant with a potent increase of the ∫ABD EMG signal within the late-E phase (*Figure 5D and F*). The expiratory airflow reached significance ($\alpha$=0.05) when comparing the second most rostral location (+0.6 mm) relative to control injections at timepoints from 4 min (KW test: $H(5)$ = 14.66, p=0.012, Dunn: p<0.001) to 12 min post-injection (KW test: $H(5)$ = 18.35, p=0.003, Dunn: p<0.001; *Figure 5D*). Moreover, airflow responses were more negative following bicuculline injections in the +0.6 mm group compared to the –0.2 mm group, and remained different at several timepoints from 8 min (KW test: $H(4)$ = 11.41, p=0.022, Dunn: p<0.001) to the last 20 min of the post-injection period (KW test: $H(4)$ = 10.65, p=0.031, Dunn: p=0.002; *Figure 5D*). In a similar manner, increases in the ∫ABD EMG at the two most rostral locations (+0.6 purple, +0.8 mm: orange) differed relative to those elicited by control injections and at the most caudal injection site from 8 min to 14 min post-injection (–0.2 mm: red) (KW test: $H(5)$ = 20.81, p<0.001, Dunn: +0.6 mm: p<0.001; +0.8 mm: p<0.001; *Figure 5F*) with the +0.8 mm group remaining elevated at 16 min (KW test: $H(4)$ = 13.56, p=0.009, Dunn: +0.8: p=0.002; *Figure 5F*). Thus, our results in the late-E phase demonstrate that the rostral regions of the ventral medulla are the most sensitive to bicuculline-induced increases in expiratory airflow, presumably elicited via a prominent activation of ABD activity.

For the inspiratory phase, bicuculline injections evoked strong increases in the respiratory airflow, which were strongest in the most rostral locations (+0.6 mm, +0.8 mm) and relatively weaker in more caudal locations (–0.2 mm). Inspiratory airflow in the +0.6 location increased relative to control injections for timepoints extending from 2 min (KW test: $H(4)$ = 13.25, p=0.021, Dunn: p<0.001) to 12 min post-injection (KW test: $H(4)$ = 14.58, p=0.012, Dunn: p=0.001; *Figure 5D*). Furthermore, we also observed increases in the airflow of the +0.6 mm location when compared to the +0.4 mm group, from 10 min (KW test: $H(4)$ = 10.50, p=0.033, Dunn: p=0.004) to 12 min (KW test: $H(5)$ = 10.93, p=0.027, Dunn: p=0.003) post-injection (*Figure 5D*). Contributions of the ABD muscle to the respiratory changes during inspiration (*Figure 5F*, inspiratory phase) were determined to be negligible as the raw ABD EMG is not active during this phase and the remaining signal in the ∫ABD EMG can be accounted for by exponential decay induced by the integration algorithm used to pre-process these signals (see Methods section). These results illustrate that the increases in inspiratory airflow evoked by bicuculline injections are stronger when targeting more rostral injection locations. However, these changes are relatively weak and, on average, remain altered for a shorter period of time relative to the effects noted above for the late-E phase.

Negative deflections in the airflow signal during the post-I phase were observed in several responses and were most pronounced in the two most rostral injection groups consistent with the results above (purple and orange, *Figure 5D*). Bicuculline injections into the most rostral group (+0.8 mm: orange) evoked substantial increases in ∫ABD EMG activity during the post-I phase which were elevated during the 6 min (KW test: $H(4)$ = 10.37, p=0.035, Dunn: p=0.002) and 16 min time bins (KW test: $H(4)$ = 9.52, p=0.049, Dunn: p=0.001) when compared to injections into the most caudal group (–0.2 mm). These effects were unlikely to be related to any changes in the DIA signal, as the DIA muscle activity was quiescent during the post-I period (*Figure 5A*) and the ∫DIA EMG is contaminated by the pre-processing as noted above for the ABD muscle during inspiration (see Methods section).

Altogether, the AUC analysis in *Figure 5* shows that airflow is substantially altered in all three phases of the respiratory cycle and the ABD muscle is strongly activated in several groups in the late-E and post-I periods with injections of bicuculline along the pFL axis. Importantly, these effects are stronger when targeting the rostral areas of the pFL (+0.6 mm and +0.8 mm) compared to injections that target the more caudal regions.

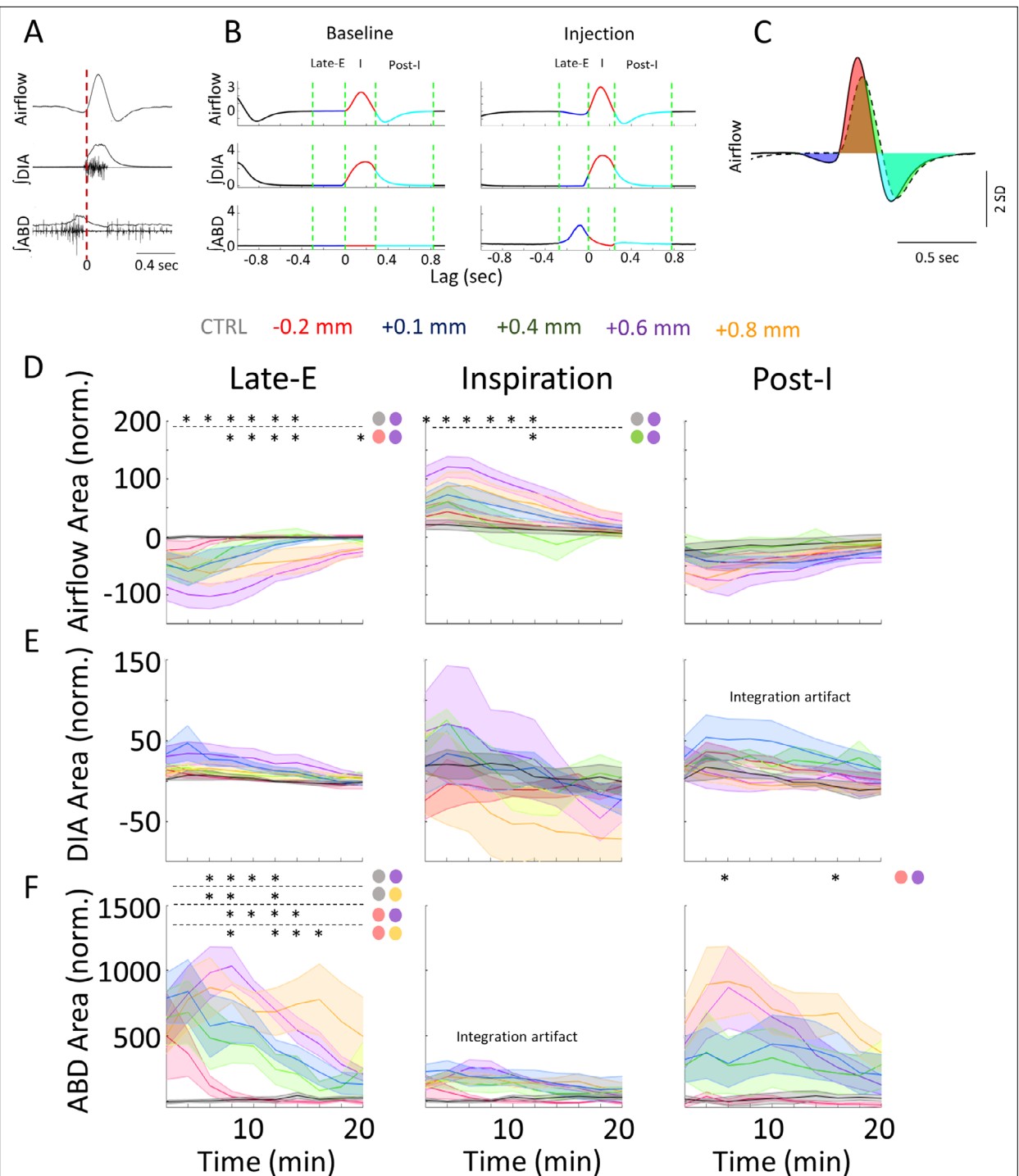

**Figure 5.** Rostral injections elicit more prominent changes to respiration in each signal and sub-period. (**A-C**) Is the same as *Figure 1*, (*Figure 1—source data 1*), it has been included here for further clarity when analyzing the results. (**A**) Raw and integrated EMG activity relative to the onset of inspiration (vertical line). (**B**) Representative mean-cycles during the baseline and post-injection, defining the late-expiratory (late-E) (blue), inspiratory (red), and post-inspiratory (post-I) (cyan) periods. (**C**) Sample calculation of normalized area during each phase. (**D–F**) Normalized response area across post-injection time for the airflow (**D**), ∫DIA EMG (**E**), and ∫ABD EMG (**F**) signals following bicuculline injections into various coordinates along the rostral-caudal axis (colors as per label above figure). Sample size for plots on D–F are as follows: CTRL, *n*=7; –0.2 mm, *n*=5; +0.1 mm, *n*=7; +0.4 mm, *n*=5; +0.6 mm, *n*=6; +0.8 mm, *n*=5. Stars indicate a significant difference ($p<0.05$) between the responses elicited by different injection locations (color coded based on the groups defined in *Figure 2B*) as assessed via Kruskal-Wallis test. Colored circles indicate the comparisons which were significant as per Dunn's post hoc test with Sidak's correction. Shaded areas indicate mean + SEM. ABD, abdominal; DIA, diaphragm; EMG, electromyogram.

*Figure 5 continued on next page*

*Figure 5 continued*

The online version of this article includes the following source data for figure 5:

**Source data 1.** Rostral injections elicit more prominent changes to respiration in each signal and sub-period, dataset for *Figure 5D–F*.

## Bicuculline responses can be differentiated by the deviations of their 3D trajectories at different phases of the respiratory cycle

We next sought to determine whether responses to bicuculline injections across the rostro-caudal axis of the pFL could be more precisely differentiated from each other when combining the responses in all three recorded signals (airflow, together with ∫DIA and ∫ABD EMG) evoked within the periods of late-E, inspiration, and post-I. Our approach was to represent all three signals as axes in a 3D space wherein each point corresponds to an individual timepoint during our recordings. When plotting the measurements across an entire respiratory cycle, this multidimensional representation resembles a looped trajectory as shown in *Figure 1D* for cycles recorded during the baseline and post-injection periods. Indeed, each phase of the cycle (late-E, inspiration, and post-I) could be recognized as a particular region along the 3D trajectory. Inspiration, for example, occupies the space where airflow and ∫DIA EMG are positive (*Figure 1D*: red). In baseline conditions, the loop is largely confined to the plane defined by the airflow-∫DIA axes, as the ABD muscles are not activated during resting states (*Figure 1E*, left). Following bicuculline injections, respiratory loops became more complex, involving marked perturbations occurring in this first plane, together with novel deformations extending along the ∫ABD EMG axis as shown in the right-hand plane in *Figure 1E*, described by airflow and the ∫ABD EMG. The relative amplitudes of these complex deviations from the baseline cycle could then be used to characterize the degree of change caused by our injections across pFL sites.

*Figure 6A–E* demonstrates how the trajectories computed during the baseline (shown in black) are altered as the response evolves across each time bin spanning the full 20 min post-injection period for all injection locations. Animations of each example recording in *Figure 6*, along with their cycle-by-cycle variability, further clarify their 3D shape as the trajectory is rotated about the ∫DIA EMG axis (*Figure 6—animations 1–5*). We noted several key deformations to baseline loops including a 'bulb' of activation on the outermost edge of the loop during the peak of inspiration when airflow is at its maximum. These 'bulbs' are similarly confined to the ∫DIA EMG-airflow axis as both of these signals are involved in shaping their overall prominence. Baseline loops are also perturbed during the late-E period as a result of ∫ABD EMG-airflow activation which can be seen as 'tails' which extend upward in the right-most plots of *Figure 6A–E*. ABD-airflow activation during the post-I period manifests as an upward 'foot-like' extension of the cycle in left-most portion of the *Figure 6A–E*, most prominently displayed in the representative recording of *Figure 6D*. The example trajectories in *Figure 6A–E* demonstrate how the deformations in each phase can be used as features to distinguish the responses of each injection location.

Consistent with our analyses above, the late-E tails of the response loops elicited by caudal injection locations are relatively smaller compared to those observed in rostral locations (*Figure 6A*). Moreover, these 'tails' in the –0.2 mm, +0.1 mm, and +0.4 mm groups last for a shorter duration than the two most rostral locations as can be determined by their more blue-yellow color (*Figure 6A–E*: right-most panel). Inspiratory bulbs were more similar across groups and were most prominent in the +0.6 mm group (*Figure 6A–E*: middle panel). Lastly, post-I 'feet' are largely absent in the example responses of the three most caudal groups (–0.2 mm, +0.1 mm, +0.4 mm) and are particularly noticeable in the +0.6 mm and +0.8 mm groups.

Quantifying the extent of these three features as the Euclidean distance between bicuculline injection and baseline in each respiratory phases is displayed in *Figure 6F*. Compared to control injections of HEPES buffer, bicuculline evoked pronounced late-E 'tails', as well as inspiratory 'bulbs', and post-I 'feet' in all injection groups which lasted throughout the 20 min response, with the notable exception of the most caudal group (–0.2 mm) which returned to a baseline trajectory by 8 min post-injection (*Figure 6G*). Consistent with the results in *Figure 5*, the most rostral injections at +0.6 mm and +0.8 mm elicited the largest late-E deformations. The 3D approach applied here further differentiated the responses of these two groups, as the late-E 'tails' of the +0.6 mm group were longer than those observed in the –0.2 mm group early in the response from 8 min (KW test: $H(4) = 12.22$, p=0.016, Dunn: p<0.001) to 12 min (KW test: $H(4) = 12.34$, p=0.015, Dunn: p=0.002), while those

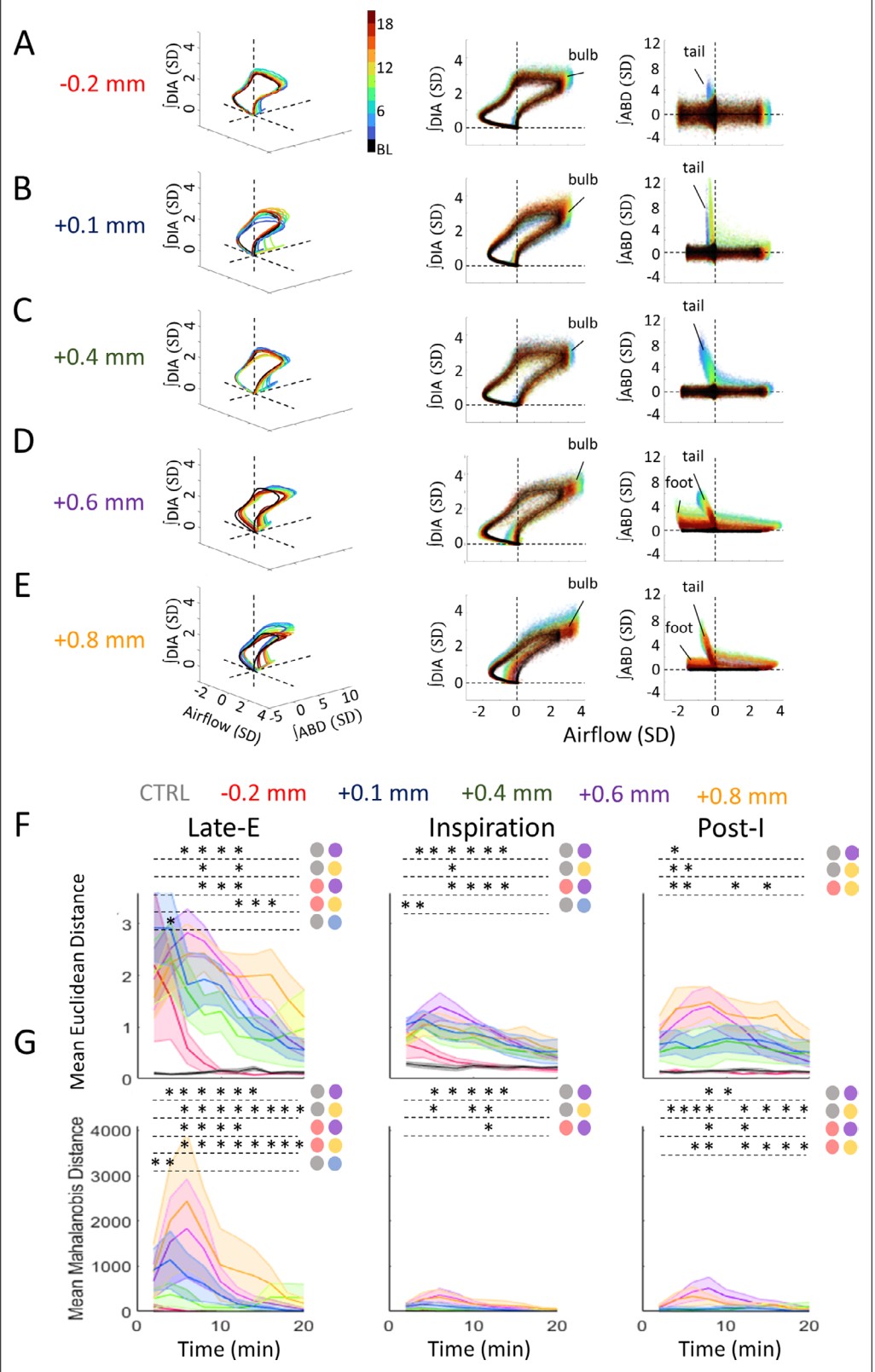

**Figure 6.** Deformations in multivariate respiratory trajectories differentiate the responses of various rostro-caudal coordinates following bicuculline injections. (**A–E**) Representative respiratory trajectories from each injection location. Each row shows the 3D trajectories (solid lines) of the mean-cycle computed for baseline (black) and response time bins (blue to red), based on the color legend in the 3D panel of A. 2D projections of these

*Figure 6 continued on next page*

*Figure 6 continued*

trajectories onto the airflow-∫DIA EMG (middle), and airflow-∫ABD EMG planes (right) follow on the right of each 3D figure. Clouds of colored points surrounding each trajectory indicate values from individual cycles of the corresponding time bin. Dashed lines indicate the origin of each axis. (**F**) Mean Euclidean distance across response time bins comparing each injection location for the late-expiratory (late-E), inspiratory, and post-inspiratory (post-I) phases. Colors as per *Figure 5*, noted above the figure. Shaded areas indicate mean ± SEM. Stars indicate significance (p<0.05) for a given time bin between injection locations or control injections as determined by Kruskal-Wallis test. Colored circles indicate which groups were significantly different as per Dunn's post hoc testing with Sidak's correction. (**G**) Mean Mahalanobis distance across the post-injection response for each injection location and respiratory phase. Colors as in F. Sample size for plots on F–G are as follows: CTRL, *n*=7; –0.2 mm, *n*=5; +0.1 mm, *n*=7; +0.4 mm, *n*=5; +0.6 mm, *n*=6; +0.8 mm, *n*=5. Shaded areas indicate mean ± SEM. Stars indicate significance at the p=0.05 level between injection locations for a given time bin as determined by Kruskal-Wallis test. Colored circles indicate which groups were significantly different as per Dunn's post hoc with Sidak's correction. ABD, abdominal; DIA, diaphragm; EMG, electromyogram.

The online version of this article includes the following video and source data for figure 6:

**Source data 1.** Deformations in multivariate respiratory trajectories differentiate the responses of various rostro-caudal coordinates following bicuculline injections.

**Figure 6—animation 1.** Rotating 3D respiratory trajectories highlight the features of breathing in baseline and response conditions.

**Figure 6—animation 2.** Rotating 3D respiratory trajectories highlight the features of breathing in baseline and response conditions.

**Figure 6—animation 3.** Rotating 3D respiratory trajectories highlight the features of breathing in baseline and response conditions.

**Figure 6—animation 4.** Rotating 3D respiratory trajectories highlight the features of breathing in baseline and response conditions.

**Figure 6—animation 5.** Rotating 3D respiratory trajectories highlight the features of breathing in baseline and response conditions.

in the +0.8 mm group peaked later in the response and were longer than in the –0.2 mm group from 12 min (KW test: $H(4) = 12.34$, p=0.015, Dunn: p=0.001) to 16 min (KW test: $H(4) = 10.89$, p=0.028, Dunn: p=0.001; *Figure 6G*). These results are consistent with the observations in *Figure 3F*, which indicate that responses in the +0.6 mm group begin earlier relative to other groups. Moreover, the +0.6 mm group produced the largest deformations to the inspiratory 'bulb' which were greater than those in the –0.2 mm group from 8 min (KW test: $H(4) = 12.59$, p=0.013, Dunn: p<0.001) to 14 min (KW test: $H(4) = 10.27$, p=0.036, Dunn: p=0.001; *Figure 6G*). On the contrary, the largest post-I 'feet' were observed in the +0.8 mm group and were greater than the –0.2 mm group from 4 min (KW test: $H(4) = 10.01$, p=0.040, Dunn: p=0.002) to 16 min (KW test: $H(4) = 9.70$, p=0.046, Dunn: p=0.001), excluding at the 10 min (KW test: $H(4) = 8.93$, p=0.063) and 14 min time bins (KW test: $H(4) = 8.10$, p=0.088; *Figure 6G*).

As demonstrated in the representative graphs in *Figure 6A–E*, the variability of the respiratory trajectories within the baseline or response periods (shown as clouds around the trajectory 'loop') differs between experiments. To account for the impact of this variability on our quantification of trajectory deformations, we computed the mean Mahalanobis distance within the late-E, inspiratory, and post-I phases (see Methods). Our results revealed a gradient of response intensities relative to respiratory variability, with the two most rostral groups producing the strongest deformations to breathing trajectories (*Figure 6H*). Similar to our results above, these groups differed in the time course and relative strength of their responses in each phase. Injections at +0.6 mm produced more pronounced late-E 'tails' compared to caudal injection responses from 6 min (KW test: $H(4) = 11.80$, p=0.019, Dunn: p=0.001) for 12 min (KW test: $H(4) = 12.24$, p=0.016, Dunn: p=0.001; *Figure 6H*). Late-E tails' from the +0.8 mm group were more prominent than 'tails' evoked by caudal bicuculline injections for much greater proportion of the 20 min response compared to the +0.6 mm group, ranging from 6 min (KW test: $H(4) = 11.80$, p=0.019, Dunn: 0.002) to 20 min post-injection (KW test: $H(4) = 11.01$, p=0.026 Dunn: p<0.001; *Figure 6H*). Deformations to the inspiratory 'bulb' relative to respiratory variability were larger in the +0.6 mm group compared to caudal responses and peaked at the 12 min time bin post-injection (KW test: $H(4) = 10.63$, p=0.031, Dunn: p=0.002) (*Figure 6H*). In

contrast, the extent of post-I 'feet' was greater in the +0.8 mm group than the –0.2 mm group from 6 min (KW test: $H(4) = 11.00$, p=0.027, Dunn: p=0.002) to 20 min (KW test: $H(4) = 10.74$, p=0.030, Dunn: p<0.001), whereas the extent of post-I 'feet' in the +0.6 mm group was greater than in caudal injections only at the 8 min (KW test: $H(4) = 11.65$, p=0.020, Dunn: p=0.002) and 12 min time bins (KW test: $H(4) = 12.23$, p=0.016, Dunn: p=0.001). Thus, while response intensity in all respiratory periods was strongest in the two most rostral injection groups, overall changes in the late-E and post-I period were strongest in the +0.8 mm group, while changes in inspiration were strongest in the +0.6 mm group.

## Discussion

In this study, our primary aim was to investigate and quantify the extent of the rostro-caudal distribution of the source of active expiration within the pFL. To accomplish this objective, we used bicuculline, a GABA-A receptor antagonist known for its ability to disinhibit pFL cells and generate active expiration (*de Britto and Moraes, 2017*; *Huckstepp et al., 2015*; *Pagliardini et al., 2011*; *Silva et al., 2019*). Bicuculline was administered at various rostro-caudal coordinates, and we methodically assessed the resulting respiratory responses to pinpoint the location within this region that induced the most significant and relevant changes to the respiratory cycle. Our findings offer valuable insights into the neural circuitry governing active expiration in the brainstem, with a particular focus on less-explored, more rostral areas of the pFL.

### Technical considerations

Previous work has already demonstrated a temporal correlation between the onset of late-E neuron activity in the pFL and ABD activity recruitment in response to bicuculline (*Pagliardini et al., 2011*; *de Britto and Moraes, 2017*; *Magalhães et al., 2021*), as well as the presence of GABAergic sIPSCs in late-E neurons (*Magalhães et al., 2021*). Since bicuculline has previously shown to exhibit inhibitory effects on $Ca^{2+}$-activated $K^+$ currents inducing nonspecific potentiation of NMDA currents (*Johnson and Seutin, 1997*), caution should be warranted in attributing our findings solely to the effects on GABAergic disinhibition as activation of late-E neurons and active expiration may also be triggered by these secondary pharmacological effects of bicuculline.

Our experiments focused on determining the area in the pFL that is most effective in generating active expiration as measured by ABD EMG activity and expiratory flow. We did not attempt in recording single-cell neuronal activity at various locations as previously shown in other studies (*Pagliardini et al., 2011*; *Magalhães et al., 2021*) as this approach would most likely find some late-E neurons across the pFL and thus not effectively discriminate between areas of the pFL. Future studies involving multi-unit recordings or imaging of cell population activities will help to determine the firing pattern and population density of bicuculline-activated cells and further determine differences in distribution and function of late-E neurons across the region of the pFL.

### Histological analysis

We adopted a stereological approach to identify the central region of the injection sites by visualizing sections containing fluorobeads. Experimental subjects were categorized based on the proximity of the injection sites, spanning from –0.2 mm to +0.8 mm from VIIc. We took measures to ensure that injections were positioned ventrolateral to the facial nucleus (VIIc) and did not overlap with PHOX2B+ cells within the more ventromedial RTN. The assessment of cellular activity, quantified through cFos staining, unveiled the existence of basal activity in control rats, that is likely emanating from subthreshold physiological processes within the pFL which do not culminate in ABD activity. Analysis of the cFos staining confirmed focal activation of neurons in the pFL of rats injected with bicuculline and minimal cFos expression in the PHOX2B+ cells in all groups as compared to the control group. These results confirm the very limited mediolateral spread of the drug from the core site of injection and support previous findings, suggesting that the majority of PHOX2B+ cells are more ventrally located in the ventral pFL (*Huckstepp et al., 2015*) and that PHOX2B+ cell recruitment is not necessary for active expiration (*de Britto and Moraes, 2017*; *Magalhães et al., 2021*).

## Characteristics of ABD signals

At rest, respiratory activity does not present with active expiration (i.e. expiratory flow below its functional residual capacity in conjunction with expiratory-related ABD muscle recruitment) and expiratory flow occurs due to passive recoil of chest wall with little to no contribution of ABD muscle activity. Active expiration and ABD recruitment can be spontaneously observed during sleep (in particular REM sleep, *Andrews and Pagliardini, 2015*; *Pisanski and Pagliardini, 2019*) and can be triggered during increased respiratory drive (e.g. hypercapnia, RTN stimulation, *Abbott et al., 2011*). Although never assessed in freely moving, unanesthetized rodents, bicuculline has been extensively used to generate active expiration and late-E neuron activity in both juvenile and adult anesthetized rats (*Pagliardini et al., 2011*; *Huckstepp et al., 2015*; *Huckstepp et al., 2015*; *Korsak et al., 2018*; *de Britto and Moraes, 2017*; *Magalhães et al., 2021*). Our efforts successfully elicited expiratory ABD responses and active expiration at all tested injection sites, consistent with previous reports (*Boutin et al., 2017*; *de Britto et al., 2020*; *de Britto and Moraes, 2017*; *Huckstepp et al., 2015*; *Korsak et al., 2018*; *Magalhães et al., 2021*; *Pagliardini et al., 2011*; *Silva et al., 2019*; *Zoccal et al., 2018*). Previous work has primarily examined the core of pFL at locations ranging from –0.2 mm to +1.0 mm from the caudal tip of the facial nucleus, a range similar to the one explored in the current study (–0.2 mm to +0.8 mm from VIIc). However, it is important to note that those studies have traditionally used the presence or absence of active expiration in response to a stimulus as the only measure to assess active expiration. Our study reveals that all locations within that range in which injections were made exhibited active expiration with various intensities. Consequently, we employed a combination of diverse respiratory measures to assess the injection site that induced the most robust respiratory effect.

Remarkably, the ABD response was consistently generated at locations rostral to VIIc, aligning with previous observations that did not find a reliable triggering of the ABD response at caudal positions (*Pagliardini et al., 2011*). This response was characterized by a late-E component at all rostro-caudal locations and, in the most rostral groups, an additional post-I component that was absent in the injections at more caudal locations. Although the configuration of the ABD response, whether late-E only, post-I, or a combination of the two, has not been extensively discussed in the literature, these distinctions may provide valuable insights into the involvement of distinct neuronal circuits within the pFL and in connection with other structures of the respiratory network in mediating the ABD response. For example, sample traces from the injection of bicuculline/strychnine from 0 mm to +1.0 mm from VIIc with 4–5% $CO_2$ exposure exhibited a tonic ABD response throughout the entire expiratory phase (without post-I or late-E peaks) (*Silva et al., 2019*), whereas sample traces from injections spanning from –0.2 mm to +0.4 mm from VIIc seemed to indicate a combination of tonic and late-E ABD responses (*de Britto and Moraes, 2017*; *Pagliardini et al., 2011*). Interestingly, the latter studies also observed tonic increases in ABD EMG (without a late-E peak), but only during baseline conditions and not in response to drug injection. In our study, we also observed a post-I ABD activation that only appeared at the most rostral locations. This post-I peak was previously observed in neonate rats during ABD activation occurring throughout apneic periods induced by systemic fentanyl injection (*Janczewski et al., 2002*). A similar biphasic pattern (late-E/post-I) was also observed in facial nerve recordings of in vitro preparations and associated with neuronal activity in the pFL (*Onimaru and Homma, 2003*; *Onimaru et al., 2006*). Therefore, although it has previously been described, the exact mechanism by which this post-I activity in the ABD muscles is generated is currently unknown . For example, the interplay between the rostral pFL and brainstem structures generating post-I activity, such as the proposed post-I oscillator (*Anderson et al., 2016*) or pontine respiratory networks, could be reasonably involved in this process. The presence of these unique responses featured at different rostro-caudal locations suggests a complex interplay among neural populations within the ventral medulla, necessitating further investigation beyond the scope of the present study.

## Temporal dynamics of the ABD response reveal a longer-lasting and shorter latency response at rostral locations

The temporal dynamics of the ABD response exhibited significant variations along the rostro-caudal axis. Responses at rostral locations displayed prolonged durations and stronger coupling with DIA signals compared to caudal locations. Previous research employing bicuculline injections to induce active expiration have reported an increase in the coupling between ABD and DIA signals at +0.1 mm

to +0.3 mm from VIIc in juvenile rats (*de Britto and Moraes, 2017*). Similarly, we observed good coupling at locations rostral to +0.1 mm from VIIc. Such coupling has been documented in other studies as an indicator of the robustness of the ABD response, either during hypercapnia/sleep or chemogenetic modulation during sleep (*Leirão et al., 2018*; *Pisanski and Pagliardini, 2019*). Therefore, in our study, we used it as a measure of the strength of the achieved activation of active expiration. Additionally, prior studies have indicated that the duration of effects induced by bicuculine/strychnine injections ranges from 16 min to 30 min (*Pagliardini et al., 2011*; *Silva et al., 2019*). However, these studies did not specify whether there were disparities in the duration of ABD responses along the rostro-caudal axis. In our study, we have conclusively established that there are indeed variations in the robustness (as denoted by ABD/DIA coupling) and duration of the ABD responses, with the more rostral locations yielding the most potent responses in this regard. Importantly, the group with injection sites at +0.6 mm from VIIc exhibited the swiftest response onset, suggesting that this area is the most critical for the generation of active expiration, either through direct activation of the expiratory oscillator or, alternatively, for providing a strong tonic drive to late-E neurons located elsewhere.

## Bicuculline injection elicited stronger respiratory effects at rostral locations in pFL

Bicuculline injections induced significant alterations in a number of respiratory parameters, reminiscent of previous studies exploring the effects of bicuculline/strychnine injections in the ventral medulla (*de Britto and Moraes, 2017*; *Pagliardini et al., 2011*; *Silva et al., 2019*). We observed a drop in respiratory frequency concomitant with the emergence of ABD recruitment in the caudal locations, similar to what was observed in those locations through pharmacological activation of pFL (*Boutin et al., 2017*; *Pagliardini et al., 2011*). However, in the most rostral locations in our study, the respiratory frequency remained unchanged. This dichotomy in the frequency response despite the presence of ABD recruitment in both cases may suggest that the slowing of the respiratory period in the most caudal locations may be a consequence of disinhibition of the adjacent Bötzinger complex. Additional evidence to support this view is that the nadir of respiratory frequency was achieved at increasing periods of time as the injections were located increasingly more rostrally, with the most caudal injection reaching the minimum respiratory frequency at 2 min post-injection, while at the most rostral injection site, the minimum respiratory frequency was observed at 6 min.

On the other hand, $V_T$ increased by 16–29% in all the studied locations except –0.2 mm, similar to what was observed previously with activation of pFL at locations ranging from –0.2 mm to +0.4 mm (*Boutin et al., 2017*; *Pagliardini et al., 2011*). However, minute ventilation increased above baseline at the +0.6 mm location in our study, whereas it decreased at the most caudal location and remained unchanged in the rest of the sites. These results are in agreement with previous observations, where $V_E$ remained unchanged at caudal locations after bicuculline injection (*Pagliardini et al., 2011*). However, it is important to highlight the absence of studies that have measured $V_T$ and $V_E$ after eliciting active expiration at more rostral locations (*Silva et al., 2019*). These results emphasize that the rostro-caudal organization of pFL is not as simple as previously thought and particularly highlights the importance of studying more rostral locations that have previously been neglected.

## Injection of bicuculline decreased oxygen consumption at the rostral locations

In terms of metabolic consequences of bicuculline-induced respiratory changes, we observed a decrease in $V_{O_2}$ at the most rostral locations, where we observed the largest changes in $V_T$ and $V_E$ produced by bicuculline injections. These changes in $V_T$ and $V_E$ were not due to physiological needs and, therefore, could be considered a form of artificial hyperventilation that would cause a drop in arterial $pCO_2$ (*Daly and Hazzledine, 1963*) and affect the gas exchange processes in the blood and peripheral tissues to reduce $V_{O_2}$. Alternatively, the ABD recruitment induced by bicuculline might not only increase $V_T$ and $V_E$ but might also facilitate respiratory mechanics in vivo (*Bosc et al., 2010*; *Giordano, 2005*; *Haupt et al., 2012*; *Ninane et al., 1992*; *Ninane et al., 1993*), reducing the work of breathing and, subsequently, causing a reduction in $O_2$ consumption. This, however, remains to be determined.

## Rostral pFL bicuculline injections induce more prominent changes to all phases of the respiratory cycle

In this study, we introduced a novel, AUC analysis algorithm, aimed at dissecting the respiratory changes evoked within each phase of the respiratory cycle (late-E, inspiration, post-I) as a result of the locale of injections and relative to baseline conditions for each of the airflow, DIA, and ABD signals. This method confirmed that the two most rostral injection sites elicited the most robust activation of ABD activity and induced the most significant alterations in expiratory airflow during both the late-E and post-I phases. Similarly, during the inspiratory phase, rostral injections generated the most pronounced changes in inspiratory airflow and DIA activity – particularly when positioned +0.6 mm from the VIIc. These findings align with the observed increases in tidal volume depicted in *Figure 4H and I* at those specific locations. Moreover, in the post-I phase, we also observed the most prominent negative airflow deflections at the most rostral injection sites, corresponding to concurrent ABD activation during that same timeframe.

By applying this novel analytical approach, which, to the best of our knowledge, has not been previously employed in research evaluating ABD responses triggered by pFL activation (*Boutin et al., 2017*; *de Britto and Moraes, 2017*; *Huckstepp et al., 2015*; *Korsak et al., 2018*; *Magalhães et al., 2021*; *Pagliardini et al., 2011*; *Silva et al., 2019*; *Zoccal et al., 2018*), we were able to reveal disparities in the AUC of ABD and airflow signals. These disparities, which went undetected with the analyses shown in *Figure 4B–E*, were particularly evident during the late-E and post-I periods. Furthermore, the division of the response into three distinct respiratory phases enabled a more precise assessment of how targeted stimulation affects a particular phase of the respiratory cycle, which could be used in future research to provide valuable insights into the underlying neural mechanisms of respiratory control.

## Bicuculline responses can be differentiated by the time course and total extent of respiratory activation during late-E, inspiration, and post-I phases

Our results certainly indicate that bicuculline injections at various pFL locations induce a complex set of interactions between DIA and ABD muscle activation, as well as changes in airflow across the inspiratory and expiratory phases of respiration. Our final goal was to develop an algorithm which could accurately represent the complete 'framework' of respiratory activation across each recorded signal with respect to each these phases. To this end, we introduced another novel analysis method that tracked the multivariate changes occurring throughout the respiratory cycle. We revealed a number of features of the respiratory cycle including late-E 'tails', inspiratory 'bulbs', and post-I 'feet' that allowed us to characterize alterations that are specifically relevant to active expiration. Given that this analysis is designed to maintain a consistent representation across conditions within experiments, these features can be catalogued into a 'framework' of the breathing cycle during both baseline and post-injection conditions. Thus, this method is both sensitive to subtle changes in the activations of each component signal, but also more importantly, it is highly robust to the variability inherent in the acquisition of respiratory physiological measurements in vivo.

By quantifying the respiratory cycle in terms of 3D measures of distance, we discriminated between the rostral and caudally evoked responses, as well as between the two most rostral locations which were previously undetected in the standard analyses. Our findings were consistent with the other analyses applied here with regard to the larger extent of respiratory loop distortions in the two most rostral positions in comparison to the more caudal injections. Furthermore, it was observed that injections in the most rostral group produced the highest amplitude changes to the late-E 'tails', as well as the longest late-E 'tails' and post-I 'feet'. Conversely, the most prominent inspiratory 'bulbs' were observed in the +0.6 mm group, aligning with the most substantial changes in $V_T$ observed in our single dimension analysis.

Collectively, these results imply that while the combined effects of bicuculline injection on the respiratory cycle were most pronounced at the two most rostral positions, there were slight variations in the effects exhibited during each respiratory phase at this level of the ventral medulla. These nuanced effects may suggest differences in the functional characteristics, phenotypes, and projections of cell populations at these specific coordinates. It is worth noting that the pFL neurons responsible for active expiration are typically quiescent in adults at rest (*de Britto and Moraes, 2017*; *Magalhães et al.,*

*2021*; *Pagliardini et al., 2011*), making their phenotypic and functional characterization challenging. Recent work started investigating electrophysiological characteristics of late-E neurons (*Magalhães et al., 2021*). However, these observations were made in neurons situated closer to the tip of the facial nucleus in juvenile rats. It will be important to investigate now whether phenotypic and electrical properties of neurons at more rostral positions share the same identity.

## pFL or RTN?

An ongoing debate centers on whether the neurons responsible for the generation of ABD recruitment are an independent oscillator composed of PHOX2B-negative, late-E firing neurons (pFL) (*de Britto and Moraes, 2017*; *Magalhães et al., 2021*; *Pagliardini et al., 2011*; *Pisanski and Pagliardini, 2019*) or part of the PHOX2B+/NMB+ chemosensitive neurons of the RTN/ventral parafacial region (*Abbott et al., 2011*; *Souza et al., 2020*; *Marina et al., 2010*). While there is evidence that specific stimulation of PHOX2B+/NMB+ neurons causes ABD activity and active expiration (*Abbott et al., 2011*; *Souza et al., 2020*), these studies have not shown recordings from late-E PHOX2B/NMB+ neurons in their investigations. However, there is concrete evidence of the existence of PHOX2B-negative late-E firing neurons that are activated in response to bicuculline injections in the ventral medulla (*de Britto and Moraes, 2017*; *Magalhães et al., 2021*; *Pagliardini et al., 2011*). Because our anatomical results show very little PHOX2B/cFos+ staining at injections sites rostral to +0.1 mm is fair to conclude that the ABD activity generated at that level is likely not the direct result of RTN activation or driven by PHOX2B+ neurons.

## Conclusion

The results of this study highlight the functional diversity of the lateral parafacial region and emphasize the strong role of the most rostral locations in generating active expiration. The use of our novel multidimensional map to assess physiological responses highlighted the functional nuances in respiratory responses to bicuculline along the pFL rostro-caudal axis, and advanced our understanding of the parafacial region.

## Additional information

### Funding

| Funder | Grant reference number | Author |
|---|---|---|
| Canadian Institutes of Health Research | 388717 | Silvia Pagliardini |
| Natural Sciences and Engineering Research Council of Canada | 2021-02926 | Clayton T Dickson |
| Natural Sciences and Engineering Research Council of Canada | | Annette Pisanski |

The funders had no role in study design, data collection and interpretation, or the decision to submit the work for publication.

### Author contributions

Annette Pisanski, Conceptualization, Data curation, Formal analysis, Investigation, Visualization, Methodology, Writing – original draft, Writing – review and editing; Mitchell Prostebby, Data curation, Software, Formal analysis, Visualization, Methodology, Writing – original draft, Writing – review and editing; Clayton T Dickson, Software, Formal analysis, Supervision, Writing – original draft, Writing – review and editing; Silvia Pagliardini, Conceptualization, Resources, Supervision, Funding acquisition, Investigation, Methodology, Writing – original draft, Project administration, Writing – review and editing

### Author ORCIDs

Annette Pisanski http://orcid.org/0000-0003-0627-1664

Mitchell Prostebby [ID] http://orcid.org/0009-0009-2017-129X
Clayton T Dickson [ID] http://orcid.org/0000-0002-3849-8110
Silvia Pagliardini [ID] http://orcid.org/0000-0002-1482-9173

### Ethics

All experimental procedures followed the guidelines set by the Canadian Council of Animal Care and received approval from the Animal Care and Use Committee (ACUC) of the University of Alberta (AUP#461).

Reviewer #1 (Public Review): https://doi.org/10.7554/eLife.94276.3.sa1
Reviewer #3 (Public Review): https://doi.org/10.7554/eLife.94276.3.sa2
Author response https://doi.org/10.7554/eLife.94276.sa3

## Additional files

### Supplementary files

• MDAR checklist

### Data availability

All data generated or analysed during this study are included in the manuscript and supporting files. The Matlab scripts used for the multivariate analysis are available on GitHub (copy archived at *Pisanski et al., 2024*).

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
