## [Editor Report · eLife assessment]

This manuscript presents experiments that address the question of whether the lateral parafacial area (pFL) is active in controlling active expiration, which is particularly significant in patient populations that rely on active exhalation to maintain breathing (eg, COPD, ALS, muscular dystrophy). This study presents **solid** evidence for a **valuable** finding of pharmacological mapping of the core medullary region that contributes to active expiration and addresses the question of where these regions lie anatomically. Results from these experiments will be of value to those interested in the neural control of breathing and other neuroscientists as a framework for how to perform pharmacological mapping experiments in the future.

---

## [Referee Report · Reviewer #1 (Public Review)]

The main focus of the current study is to identify the anatomical core of an expiratory oscillator in the medulla using pharmacological disinhibition. Although expiration is passive in normal eupneic conditions, activation of the parafacial (pFL) region is believed to evoke active expiration in conditions of elevated ventilatory demands. The authors and others in the field have previously attempted to map this region using pharmacological, optogenetic and chemogenetic approaches, which present with their own challenges.

In the present study, the authors take a systematic approach to determine the precise anatomical location within the ventral medulla's rostro-caudal axis where the expiratory oscillator is located. The authors used a bicuculline (a GABA-A receptor antagonist) and fluorobeads solution at 5 distinct anatomical locations to study the effects on neuronal excitability and functional circuitry in the pFL. The effects of bicuculline on different phases of the respiratory cycle were characterized using a multidimensional cycle-by-cycle analysis. This analysis involved measuring the differences in airflow, diaphragm electromyography (EMG), and abdominal EMG signals, as well as using a phase-plane analysis to analyze the combined differences of these respiratory signals. Anatomical immunostaining techniques were also used to complement the functional mapping of the pFL.

Major strengths of this work include a robust study design, complementary neurophysiological and immunohistochemical methods and the use of a novel phase-plane analysis. The authors construct a comprehensive functional map revealing functional nuances in respiratory responses to bicuculline along the rostrocaudal axis of the parafacial region. They convincingly show that although bicuculline injections at all coordinates of the pFL generated an expiratory response, the most rostral locations in the lateral parafacial region play the strongest role in generating active expiration. These were characterized by a strong impact on the duration and strength of ABD activation, and a robust change in tidal volume and minute ventilation. The authors also confirmed histologically that none of the injection sites overlapped grossly with PHOX2B+ neurons, thus confirming the specificity of the injections in the pFL and not the neighboring RTN.

Although a caveat of the approach is that bicuculine injections have indiscriminate effects on other neuronal populations in the region (GABAergic, glycinergic, and glutamatergic), the results can largely be interpreted as modulation of neuronal populations in different regions of the pFL have differential effects on expiratory output. This limitation of the pharmacological approach has also been aptly discussed by the authors.

Collectively, these findings advance our understanding of the presumed expiratory oscillator, the pFL, and highlight the functional heterogeneity in the functional response of this anatomical structure.

---

## [Referee Report · Reviewer #3 (Public Review)]

Summary:

The study conducted by Pisanski et al investigates the role of the lateral parafacial area (pFL) in controlling active expiration. Stereotactic injections of bicuculline were utilized to map various pFL sites and their impact on respiration. The results indicate that injections at more rostral pFL locations induce the most robust changes in tidal volume, minute ventilation, and combined respiratory responses. The study indicates that the rostro-caudal organization of the pFL and its influence on breathing is not simple and uniform.

Strengths:

The data provide novel insights into the importance of rostral locations in controlling active expiration. The authors use innovative analytic methods to characterize the respiratory effects of bicuculline injections into various areas of the pFL.

Weaknesses:

Bicuculline injections increase the excitability of neurons. Aside of blocking GABA receptors, bicuculline also inhibits calcium-activated potassium currents and potentiates NMDA currents, thus insights into the role of GABAergic inhibition are limited.

Increasing the excitability of neurons provides little insights into the activity pattern and function of the activated neurons. Without recording from the activated neurons, it is impossible to know whether an effect on active expiration or any other respiratory phase is caused by bicuculline acting on rhythmogenic neurons or tonic neurons that modulate respiration. While this approach is inappropriate to study the functional extent of the conditional "oscillator" for active expiration, it still provides valuable insights into this region's complex role in controlling breathing .

---

## [Author Response]

The following is the authors’ response to the original reviews.

**eLife assessment**
This manuscript presents a solid and generally convincing set of experiments to address the question of whether the lateral parafacial area (pFL) is active in controlling active expiration, which is particularly important in patient populations that rely on active exhalation to maintain breathing (eg, COPD, ALS, muscular dystrophy). This study presents a valuable finding by pharmacologically mapping the core medullary region that contributes to active expiration and addresses the question of where these regions lie anatomically. Results from these experiments will be of value to those interested in the neural control of breathing and other neuroscientists as a framework for how to perform pharmacological mapping experiments in the future.

Thanks for the positive feedback on our study, as well as the assessment of the novelty of our investigation and the advancements to the field that these results will bring in the future.

We have addressed the specific comments and made changes to the manuscript as indicated below.

**Public Reviews:**

**Reviewer #1 (Public Review):**
The main focus of the current study is to identify the anatomical core of an expiratory oscillator in the medulla using pharmacological disinhibition. Although expiration is passive in normal eupneic conditions, activation of the parafacial (pFL) region is believed to evoke active expiration in conditions of elevated ventilatory demands. The authors and others in the field have previously attempted to map this region using pharmacological, optogenetic, and chemogenetic approaches, which present their own challenges.In the present study, the authors take a systematic approach to determine the precise anatomical location within the ventral medulla's rostrocaudal axis where the expiratory oscillator is located. The authors used a bicuculline (a GABA-A receptor antagonist) and fluorobeads solution at 5 distinct anatomical locations to study the effects on neuronal excitability and functional circuitry in the pFL. The effects of bicuculline on different phases of the respiratory cycle were characterized using a multidimensional cycle-by-cycle analysis. This analysis involved measuring the differences in airflow, diaphragm electromyography (EMG), and abdominal EMG signals, as well as using a phase-plane analysis to analyze the combined differences of these respiratory signals. Anatomical immunostaining techniques were also used to complement the functional mapping of the pFL.Major strengths of this work include a robust study design, complementary neurophysiological and immunohistochemical methods, and the use of a novel phase-plane analysis. The authors construct a comprehensive functional map revealing functional nuances in respiratory responses to bicuculline along the rostrocaudal axis of the parafacial region. They convincingly show that although bicuculline injections at all coordinates of the pFL generated an expiratory response, the most rostral locations in the lateral parafacial region play the strongest role in generating active expiration. These were characterized by a strong impact on the duration and strength of ABD activation and a robust change in tidal volume and minute ventilation. The authors also confirmed histologically that none of the injection sites overlapped grossly with PHOX2B+ neurons, thus confirming the specificity of the injections in the pFL and not the neighboring RTN.Collectively, these findings advance our understanding of the presumed expiratory oscillator, the pFL, and highlight the functional heterogeneity in the functional response of this anatomical structure.

Thanks for the positive feedback on the results presented in the current manuscript.

**Reviewer #2 (Public Review):**
Summary:Pisanski and colleagues map regions of the brainstem that produce the rhythm for active expiratory breathing movements and influence their motor patterns. While the neural origins of inspiration are very well understood, the neural bases for expiration lag considerably. The problem is important and new knowledge pertaining to the neural origins of expiration is welcome.The authors perturb the parafacial lateral (pFL) respiratory group of the brainstem with microinjection of bicuculline, to elucidate how disinhibition in specific locations of the pFL influences active expiration (and breathing in general) in anesthetized rats. They provide valuable, if not definitive, evidence that the borders of the pFL appear to extend more rostrally than previously appreciated. Prior research suggests that the expiratory pFL exists at the caudal pole of the facial cranial nucleus (VIIc). Here, the authors show that its borders probably extend as much as 1 mm rostral to VIIc. The evidence is convincing albeit with caveats.Strengths:The authors achieve their aim in terms of showing that the borders of the expiratory pFL are not well understood at present and that it (the pFL) extends more rostrally. The results support that point. The data are strong enough to cause many respiratory neurobiologists to look at the sites rostral to the VIIc for expiratory rhythmogenic neurons and characterize their properties and mechanisms. At present my view is that most respiratory neurobiologists overlook the regions rostral to VIIc in their studies of expiratory rhythm and pattern.Weaknesses:The injection of bicuculline has indiscriminate effects on excitatory and inhibitory neurons, and the parafacial region is populated by excitatory neurons that are expiratory rhythmogenic and GABA and glycinergic neurons whose roles in producing active expiration are contradictory (Flor et al. J Physiol, 2020, DOI: 10.1113/JP280243). It remains unclear how the microinjections of bicuculline differentially affect all three populations. A more selective approach would be able to disinhibit the populations separately. Nevertheless, for the main point at hand, the data do suggest that we should reconsider the borders of the expiratory pFL nucleus and begin to examine its physiology up to 1 mm rostral to VIIc.The control experiment showed that bicuculline microinjections induced cFos expression in the pFL, which is good, but again we don't know which neurons were disinhibited: glutamatergic, GABAergic, or glycinergic.

Thanks for sharing your excitement on the results of our study, and appreciating the thorough investigation performed with the use of bicuculline, an approach that was originally used in Pagliardini et al, 2011, PMID: 21414911 and then used by many other groups to generate and study active expiration in vivo.

In the current study we used the well known effect of Bicuculline to systematically test the area that is more sensitive to such a pharmacological effect, and hence may be the core for generating active expiration. While the use of GABA receptor antagonists may have an indiscriminate effect on GABA receptor expressing neurons with various phenotypes, anatomical assessment of inhibitory cells has shown very little distribution of GABAergic and glycinergic cells in the parafacial area (Tanaka et.al, 2003; PMID: 14512139) and it has been inferred in multiple publications (Huckstepp et al., 2015, PMID: 25609622; Huckstepp et al. 2016 PMID: 27300271; Huckstepp et al., 2018, PMID: 30096151; Flor et al., 2020, PMID: 32621515; Britto & Moraes, 2017; PMID: 28004411; Silva et al. 2016; PMID: 26900003) and demonstrated recently (Magalhaes et al., 2021; PMID: 34510468) that late-E neurons in the parafacial region are excitatory and have a glutamatergic phenotype. We can’t exclude that a small fraction of neurons in the pFL area are inhibitory, and that they could influence recruitment of adjacent late-E expiratory neurons. A more selective activation of neuronal populations with different phenotype would be indeed interesting, nonetheless, if local inhibitory neurons have a role in the generation of active expiration, then their disinhibition could have either an inhibitory effect on late-E activity or stimulate expiration in a more indirect fashion.

While the effect of bicuculline on active expiration has been reported and replicated in multiple manuscripts, the source of inhibition across different phases of the respiratory cycle is still under investigation. Some studies suggest that GABAergic and glycinergic inhibition is not originated in pFL but rather in the BötC and preBötC areas (Flor et al., 2020, PMID: 32621515; Magalhaes et al., 2021; PMID: 34510468) and the effects of this inhibition across the respiratory cycle is debated. Future studies will be key to identify the source of pFL inhibition.

The manuscript characterizes how bicuculline microinjections affect breathing parameters such as tidal volume, frequency, ventilation, inspiratory and expiratory time, as well as oxygen consumption. Those aspects of the manuscript are a bit tedious and sometimes overanalyzed. Plus, there was no predictive framework established at the outset for how one should expect disinhibition to affect breathing parameters. In other words, if the authors are seeking to map the pFL borders, then why analyze the breathing patterns so much? Does doing so provide more insight into the borders of pFL? I did not think it was compellingly argued.

We have edited the introduction to address this comment and emphasize the rationale for the study. We also edited the results section to summarize our findings.

We continue to report our in-depth analysis of the perturbations induced by bicuculline injection over the various respiratory characteristics as this will be fundamental to determine the effects of our experiment not only on the activation of pFL and active expiration, but also on the respiratory network in general. In order to be fair and open about our findings we have reported the results of our analysis in detail. Of note, all sites generated active expiration, but since the objective of the study was to determine the sites with the most significant changes, a finer and multilevel analysis has been used.

Further, lines 382-386 make a point about decreasing inspiratory time even though the data do not meet the statistical threshold. In lines 386-395, the reporting appears to reach significance (line 388) but not reach significance (line 389). I had trouble making sense of that disparity.

The statistics were confirmed, and the lines edited as follows: “Interestingly, the duration of inspiration during the response was found to decrease in all groups relative to baseline respiration (Ti response = 0.279 ± 0.034s, Ti baseline = 0.318 ± 0.043s, Wilcoxon rank sum: Z = 3.24, p = 0.001). Contrary to this decrease in inspiratory duration, the total expiratory time was observed to increase in all groups and remained elevated compared to baseline (TE response = 1.313 ± 0.188s, TE baseline = 1.029 ± 0.161s, Wilcoxon rank sum: Z = 4.49, p = 0.001).”

The other statistical hiccups include "tended towards significance" (line 454), "were found to only reach significance for a short portion of the response" (line 486-7), "did not reach the level of significance" (line 506), which gives one the sense of cherry picking or over-analysis. Frankly, this reviewer finds the paper much more compelling when just asking whether the microinjections evoke active expiration. If yes, then the site is probably part of the pFL.

Statistical “tendencies” have been eliminated throughout the manuscript.

We have analyzed in details our results in order to determine changes and differential effects on respiration when comparing the 5 sites of injections. Although the presentation of the results may seem tedious, it has allowed us to highlight some interesting effects: first, the effects on respiratory frequency. It has been shown in the past that optogenetic stimulation of this area causes an increase in respiratory frequency (Pagliardini et al., 2011, PMID: 21414911), whereas a dishinibition with this same approach or stimulation of AMPAreceptor in pFL have shown a reduction in frequency or not a significant change in the response (Pagliardini et al., 2011, PMID: 21414911; Huckstepp et al., 2015, PMID: 25609622; Huckstepp et al. 2016 PMID: 27300271; Huckstepp et al., 2018, PMID: 30096151). Here, we suggest that the reduction in respiratory frequency is observed only in the caudal sites and could be attributed to BötC effects rather than the stimulation of the core of the pFL since no respiratory change was observe where the effect was more potent (rostral side). Another interesting point was the effects on O2 consumption, although difficult to interpret at this point, we found very interesting that hyperventilation occurred only at the most rostral injection sites.

I encourage the authors to consider the fickleness of p-values in general and urge them to consider not just p but also effect size.

Thank you for the feedback on our description of the statistical results and the suggestion of incorporating effect size. We have now included measurements of effect size in the results section. Specifically, we calculated the effect size within each ANOVA using the value of eta squared for all data shown in Figures 3 and 4. Please note that in our phase-plane analysis (Fig. 5-6) the Mahalanobis distance is itself an effect size measure for multidimensional data. We also note that statistical evaluation using non-parametric analyses do not involve effect sizes.

**Reviewer #3 (Public Review):**
Summary:The study conducted by Pisanski et al investigates the role of the lateral parafacial area (pFL) in controlling active expiration. Stereotactic injections of bicuculline were utilized to map various pFL sites and their impact on respiration. The results indicate that injections at more rostral pFL locations induce the most robust changes in tidal volume, minute ventilation, and combined respiratory responses. The study indicates that the rostrocaudal organization of the pFL and its influence on breathing is not simple and uniform.Strengths:The data provide novel insights into the importance of rostral locations in controlling active expiration. The authors use innovative analytic methods to characterize the respiratory effects of bicuculline injections into various areas of the pFL.Weaknesses:Bicuculline injections increase the excitability of neurons. Aside from blocking GABA receptors, bicuculline also inhibits calcium-activated potassium currents and potentiates NMDA current, thus insights into the role of GABAergic inhibition are limited.Increasing the excitability of neurons provides little insights into the activity pattern and function of the activated neurons. Without recording from the activated neurons, it is impossible to know whether an effect on active expiration or any other respiratory phase is caused by bicuculline acting on rhythmogenic neurons or tonic neurons that modulate respiration. While this approach is inappropriate to study the functional extent of the conditional "oscillator" for active expiration, it provides valuable insights into this region's complex role in controlling breathing.

We have included a reflection of the weaknesses of our studies in the technical consideration section to address the possibility that bicuculline may induce active expiration through other mechanisms. Please note that the use of bicuculline was not to gain further insight on GABAergic inhibition of pFL but to adopt a tool to generate active expiration that has been extensively validated by our group and others.

Multiple studies have shown recruitment of excitatory late expiratory neurons with bicuculline injections. Although we did not record from late-E neurons in this study, we infer from the body of literature that disinhibition of neurons in this area will activate late-E neurons (as previously demonstrated) and generate active expiration. Although we see value in recording activity of single neurons (especially to study mechanisms of rhythmogenesis), we opted to measure the physiological response from respiratory muscles as an indication of active expiration recruitment *in vivo*. Recording from single neurons after bicuculline injections in each site would confirm the presence of expiratory neurons along the parafacial area, which is probably not surprising, since every site tested promoted active expiration. The focus of the study though was to determine the site with the strongest physiological response to disinhibition. Future studies will be key to determine whether all neurons along this column have similar electrophysiological rhythmic properties to the ones recently reported (Magalhaes et al., 2021; PMID: 34510468), or some of them simply provide tonic drive to late-E neurons located elsewhere.

We have discussed the issue as follows:

“Our experiments focused on determining the area in the pFL that is most effective in generating active expiration as measured by ABD EMG activity and expiratory flow. We did not attempt to record single cell neuronal activity at various locations as previously shown in other studies (Pagliardini et al 2011; Magalhaes et al., 2021), as this approach would most likely find some late-E neurons across the pFL and thus not effectively discriminate between areas of the pFL. Future studies involving multi-unit recordings or imaging of cell population activities will help to determine the firing pattern and population density of bicuculline-activated cells and further determine differences in distribution and function of late-E neurons across the region of the pFL.”

**Recommendations for the authors:**

**Reviewer #1 (Recommendations For The Authors):**
Overall, the manuscript addresses an important question in the field, the anatomical location of the expiratory oscillator. I commend the authors for a well-thought-out and clearly presented study. However, a few small concerns deserve attention to improve the clarity of the report.(1) The figures would benefit from a rostral-to-caudal representation of results instead of a caudal-to-rostral orientation. Example, Figure 2.

We opted for a caudal to rostral representation to progressively move away from the inspiratory oscillator (preBötC) and the anatomical reference point (the caudal tip of the facial nucleus) with our series of injections.

(2) A discussion about how expiratory responses generated by these pharmacological approaches would compare to endogenous baseline conditions. The authors mention that bicuculline injections elicited a late-E downward inflection that was absent in baseline conditions. Thus, this raises the point of how these findings compare to awake freely moving animals or during different conditions of increased ventilatory demand.

This is an interesting question that has not yet been address in the field. As far as we know, there are no recordings of pFL neurons in freely behaving animals although recordings of pFL late-E neurons under elevated PaCO2 have shown a late-E activity in in situ preparations (Britto & Moraes, 2017; PMID: 28004411; Magalhaes et al., 2021; PMID: 34510468).

We have clarified this in the discussion as follows:

“At rest, respiratory activity does not present with active expiration (i.e, expiratory flow below its functional residual capacity in conjunction with expiratory-related ABD muscle recruitment) and expiratory flow occurs due to passive recoil of chest wall with no contribution of abdominal activity. Active expiration and abdominal recruitment can be spontaneously observed during sleep (in particular REM sleep, Andrews and Pagliardini, 2015; Pisanski et al., 2019) and can be triggered during increased respiratory drive (e.g. Hypercapnia, RTN stimulation, Abbott et al., 2011). Although never assessed in freely moving, unanesthetized rodents, bicuculline has been extensively used to generate active expiration and late-E neuron activity in both juvenile and adult anesthetized rats (Pagliardini et al., 2011; Huckstepp et al., 2015 Huckstepp et al., 2016; Huckstepp et al., 2018; De Britto and Moraes, 2017; Magalhaes et al., 2021). “

(3) In Figure 2A, there appears to be an injection site in the top right quadrant of the image, very distant from the intended site. Could the authors confirm if this is an artifact?

Yes, it is an artifact of image acquisition, we should have marked that in the figure. To avoid confusion and follow other reviewers’ suggestions we have edited he figure.

(4) A stylistic suggestion would be to include the subpanel of Figure 2C saline control injection as a graph of its own and also include the control anatomical location in 2B.

Thanks for the suggestion. Because of the complex organization of the figure we opted to leave it as a subpanel in order to not distract the reader from the 5 injection sites, but still provide information about vehicle injection and their lack of changes in respiratory response.

(5) The authors note that DIAm Area (norm.) during the inspiratory phase is increased in the +6 and +8mm groups. However, Figure 5E shows that the +8mm group is significantly reduced as compared to the +6mm group. Please clarify.

During the inspiratory phase we did not observe any significant change in the DIA Area (norm.). We realize that the description of this part of the results was confusing and therefore we have eliminated that section.

**Reviewer #2 (Recommendations For The Authors):**
I encourage the authors to consider the fickleness of p-values in general and urge them to consider not just p but also effect size. There is a valuable editorial in this week's J Physiology (https://doi.org/10.1113/JP285575) that may provide helpful guidance.

Thanks for this comments and the general assessment. We realized that the results section was dense and with a lot of information. We significantly slimmed the description of the results in order to facilitate the appreciation of the results and avoid confounding statement about significant vs non- significant results.

We have now included measurements of effect size in the results section. Specifically, we calculated the effect size within each ANOVA using the value of eta squared for all data shown in Figures 3 and 4. Please note that in our phase-plane analysis (Fig. 5-6) the Mahalanobis distance is itself an effect size measure for multidimensional data. We also note that statistical evaluation using non-parametric analyses do not involve effect sizes.

The equipment and resources should be clearly identified and use RRIDs whenever possible. Resources like antibodies and other reagents (e.g., cryoprotectants) should be identified, not just by manufacturer, but also by specific part or product numbers or identifiers.

Manuscript has been edited to add these details.

The manuscript makes reference to ImageJ and Matlab routines, which must be public through GitHub or another stable repository.

Thanks for pointing this out. Image J analysis has been performed following scripts already available to users (no custom scripts). The Matlab scripts used for the multivariate analysis is now available at: https://github.com/mprosteb/Pisanski2024

The way that ABD-DIA coupling was assessed was unclear from the Methods.

The following text has been added to the methods: “The coupling between ABD and DIA signals was measured as a ratio and analyzed by quantifying the number of bursts of activity observed for the ABD and DIA EMG signals during the first 10 minutes of the response, excluding time bins at end of the response (due to fading and waning of the ABD response in those instances).”

Fig. 1A was never cited in the text.

It has been cited now.

Fig. 1A-C appears to be exactly the same as Fig. 5A-C.

The reviewer is correct. We have used figure 1 to describe and explain our analytical methods with sample data and Figure 5 describes our results. We have clarified that in: “Figure 5: Rostral injections elicit more prominent changes to respiration in each signal and sub-period. A-C: Is the same as Method Figure 1, has been included here for further clarity when analyzing the results.”

Late Expiratory airflow is given in units of volts (V) in lines 358-363 (Fig. 4C) but then in units of volts-seconds (V•s) in lines 363-367. Both units are problematic because the voltage is neither an air volume nor an air volume per unit time. Is there some conversion factor left out?

In this section of the results we describe the changes in expiratory peak amplitude (V) and expiratory peak flow (V•s). Since calibration of airflow was performed on the positive flow and for larger volumes, we prefer to use the original units to guarantee precise assessment of the change and avoid introducing potential errors. Since the analysis considers changes from baseline readings, converting to ml or ml*s would not affect our analysis.

**Reviewer #3 (Recommendations For The Authors):**
The study conducted by Pisanski et al investigates the role of the lateral parafacial area (pFL) in respiratory control, specifically in modulating active expiration. The precise location of this expiratory oscillator within the ventral medulla remains uncertain, with some studies indicating that the caudal tip of the facial nucleus (VIIc) forms the core while others propose more rostral areas. Bicuculline injections were utilized at various pFL sites to explore the impact of these injections on respiration. The authors use innovative and impressive analytic methods to characterize the effect on respiratory activity. The results indicate that injections at more rostral pFL locations induce the most robust changes in tidal volume, minute ventilation, and combined respiratory responses. The study will contribute to an enhanced understanding of the neural mechanisms controlling active expiration. The main message of the study is that the rostro-caudal organization of the pFL is not simple and uniform. The data provides novel insights into the importance of rostral locations in controlling active expiration (see e.g. lines 738-740).The data and results of the paper are intriguing, and it appears that the experiments are well-managed and executed. However, there are several major and minor comments and suggestions that should be addressed by the authors:(1) The study relies heavily on local injections into specific areas that are confirmed histologically. One potential concern is the injection volume of 200 nL in such a tiny area. The authors suggest that the drug did not spread to rostral/caudal areas outside the specified coordinate partly based on their cFOS staining. For example, the lack of cFOS activation in TH+ cells and Phox2B cells is interpreted as proof that bicuculline did not spread to these somas (Figure 2). The authors seem to use a similar argument as evidence that the pFL does not include Phox2B neurons in the RTN as discussed in the Discussion section (lines 830-847). However, it is very surprising that bicuculline injections into an area that is known to contain Phox2B and Th+ neurons do not activate these neurons as assessed by the cFOS staining. It seems puzzling to me that none of their injections shown in Figure 2 activated Phox2B or Th neurons. I assume that in targeting the pFL the authors must have sometimes hit areas that included neurons that define the RTN, which would have activated Phox2B or Th+ neurons. Did the authors find that these activations did not activate active expiration? Such negative "controls" would strengthen their argument that pFL is a separate and distinct region that selectively controls active expiration.

Thanks for the positive feedback on the manuscript. As it has been demonstrated and discussed in several previous publications, PHOX2B expressing neurons in this area of the brain are part of the RTN Neuromedin B positive neurons (more densely located in the ventral paraFacial rather than the lateral parafacial, our site of injection), the TH+ C1 neurons (located in a somewhat more caudal and medial position compared to our sites of injection, around the BötC/ preBötC area) and the large Facial MN (easily identifiable by their large size and compact location). Given this differential spatial distribution, and the controls described below, we believe we have reduced the possibility of the direct activation of these neurons, although we can’t exclude it in full.

There is now strong evidence about lack of PHOX2B expression in late E neuron in juvenile and adult rats (Magalhaes et al., 2021; PMID: 34510468). We realize that the microinjected solution could potentially diffuse in the brain and hit other areas, but we combined two strategies to verify our intention for a focal injection activating only a restricted area of the brain (i.e., the pFL): (i) localization of fluorobeads that were diluted in the Bicuculline solution; (ii) expression of cFos combined with anatomical markers, to identify activated cells. Fluorobeads have a very limited spread in the brain and therefore informed us of the site of the injection to differentiate between the five injections locations. Although we can’t assume that Bicuculline will have a similar spread (and it will also be quickly degraded in the tissue), the combination of this analysis with the localized expression of cFos cells has helped us to differentiate between injections site. Because of the proximity of PHOX2B cells in RTN and C1 neurons, we also combined cFos expression with immunohistochemistry to determine whether bicuculline activation was also visible in these two neuronal populations. Our results indicate that there is baseline cfos activity in RTN neurons (see vehicle injection) but the fraction of PHOX2B activated cells did not increase with bicuculline injections suggesting that these neurons were not the target of our injections. Please note that cfos expression has been extensively used to determine RTN neuron activation, especially following chemoreflex responses.

(2) The authors refer to "the expiratory oscillator" throughout the manuscript (e.g. lines 58, 62, 65) as if there is only one expiratory oscillator i.e. "the expiratory oscillator". For some reason, the authors avoided citing and mentioning PiCo (Anderson et al. 2016), which is considered the oscillator for postinspiration. Since the present study focuses on the role of expiration, and since the authors describe convincing effects on postinspiration, considering this oscillator which is located dorsomedial to the VRC seems relevant for the present study.

Due to the limited and controversial literature that is currently present describing Pico as a third oscillator and the fact that our studies do not directly assess the post-inspiratory activity (as measure by the V nerve or laryngeal muscles) or Pico activity and location (which would be even more distant than the RTN, for example), we prefer to avoid commenting on the effects of this injection on Pico or the connectivity between Pico and pFL.

We have added this to the discussion:

“Therefore, although it has previously been described, it is currently unknown the exact mechanism by which this post-I activity in the ABD muscles is generated. For example the interplay between the rostral pFL and brainstem structures generating post-inspiratory activity, such as the proposed post-inspiratory oscillator (PiCo; Anderson et al., 2016) or pontine respiratory networks, could be reasonably involved in this process.”

(3) The authors do not specify what type of bicuculline they injected. Bicuculline is known to have significant effects on potassium channels. Thus, the effects reported here could be due to a non-specific change in excitability, rather than caused by a specific GABAergic blockade.The authors also do not know what effects these injections cause in the neurons in vivo, since the injections are not accompanied by recordings from the respiratory neurons that they activate. This together with the non-specific bicuculline effects will affect the interpretation of the results. Thus, the authors need to be more careful when interpreting their effects as "GABAergic". The use of more specific blockers like gabazine could partly address this concern. The authors have to discuss this in a "limitation section".

Thanks for pointing that out, we have now clarified in the methods section that we used bicuculline methochloride. We can’t exclude that some side- effects could be present due to the use of this drug. For the purpose of this study though, we focused on using bicuculline as a tool to consistently generate active expiration since it has been extensively used by multiple laboratories to induce abdominal muscle recruitment and active expiration, as well as to directly record late-E neurons in this same area.

We have included in the discussion the following statement:

“Technical considerations

Bicuculline methiodide has previously been observed to exhibit inhibitory effects on Ca2+ activated K+ currents inducing non-specific potentiation of NMDA currents (Johnson and Seutin, 1997). Consequently, caution is warranted in attributing our findings solely to the GABAa antagonist properties of bicuculline. Previous work has demonstrated a temporal correlation between the onset of late-E neuron activity in the caudal parafacial region and ABD activity in response to bicuculline (Pagliardini et al., 2011; de Britto and Moraes, 2017; Magalhaes et al., 2021) as well as GABAergic sIPSCs in late-E neurons (Magalhaes et al., 2012). However, it is essential to note that the current study lacks single unit recording, preventing us from definitively confirming whether the observed activity stems from late-E neuronal GABAergic dishinibition or excitation through non GABAergic mechanisms.”

(4) I also caution the authors when stating that the bicuculline injections will reveal the precise location and functional boundaries of "the" expiratory oscillation within the pFL. Increasing the excitability with bicuculline is inappropriate to study the functional boundaries of an oscillator. It is particularly inappropriate to identify the boundaries of the pFL, a network that is normally inactive and activated only under certain behavioral and metabolic conditions. Because the injections are increasing the neuronal excitability unspecifically, and because the authors are not recording the activity of the neurons in the pFL region it is unclear what kind of neurons are activated. The cFOS staining may help to define whether these neurons are Phox2B or Th positive or negative, but they will not provide insights into the activity patterns of the activated neurons. Thus, it is fair to assume that these injections will likely include also tonic neurons that might indirectly control the activity of pFL neurons under certain metabolic or behavioral conditions without actually being involved in the rhythmogenesis of active expiration. Many of the effects peak after several minutes, and different regions cause differential effects with different time courses, which is difficult to interpret functionally. Thus, the "core" identified in the present study could consist of tonic neurons as opposed to rhythmic neurons generating active expiration.

We agree with the reviewer that our local injections may have activated an heterogeneous population of neurons. We do not claim that we only activated late-E rhythmogenic neurons but that our multiple sites of injections revealed the area that is generating the strongest excitation of ABD muscles and active expiration.

While the use of GABA receptor antagonists may have an indiscriminate effect on GABA receptor expressing neurons with various phenotypes, anatomical assessment of inhibitory cells has shown very little distribution of GABAergic and glycinergic cells in the parafacial area (Tanaka et.al, 2003; PMID: 14512139) and it has been inferred in multiple publications (Huckstepp et al., 2015, PMID: 25609622; Huckstepp et al. 2016 PMID: 27300271; Huckstepp et al., 2018, PMID: 30096151; Flor et al., 2020, PMID: 32621515; Britto & Moraes, 2017; PMID: 28004411; Silva et al. 2016; PMID: 26900003) and demonstrated recently (Magalhaes et al., 2021; PMID: 34510468) that late-E neurons in the parafacial region are excitatory and have a glutamatergic phenotype

As suggested by the reviewer, it is possible that the bicuculline injection may have activated some tonic non rhythmogenic neurons which could activate the expiratory oscillator located elsewhere.

We have edited the introduction as follows:

“By strategically administering localized volumes of bicuculline at multiple rostrocaudal levels of the ventral brainstem, we aimed to selectively enhance the excitability of neurons driving active expiration, thereby revealing the extension of the pharmacological response and the most efficient site in generating active expiration.”

We have edited the results as follows:

“Importantly, the group with injection sites at +0.6 mm from VIIc exhibited the swiftest response onset, suggesting that this area is the most critical for the generation of active expiration, either through direct activation of the expiratory oscillator or, alternatively, for providing a strong tonic drive to late-E neurons located elsewhere.”

In the introduction, it should also be emphasized that the pharmacological approach used in the present study complements the existing elegant chemogenetic studies, rather than emphasizing primarily the limitations of the chemogenetic inhibitions. The conclusion should be that these studies together provide different, yet complementary insights: The chemogenetic approach by inhibiting neurons, the present study by exciting neurons, and all studies come with their own limitations.

Thanks for the suggestion, we have updated the manuscript as follows:

“Although both of these elegant chemogenetic studies have contributed extensively to our understanding of the pFL, the existing evidence suggests that the expiratory oscillator may expand beyond the limits of the viral expression achieved in said studies, as proposed by Huckstepp et al., (2015).”

Throughout the manuscript, the authors have to be cautious when implying that an excitatory effect relates to the activity of rhythmogenic pFL neurons. For example, on line 710 the authors state that "it is conceivable to infer that the rostral pFL is in the closest proximity to the cells responsible for the generation of active expiration". While it may indeed be "conceivable", the bicuculline injections themselves provide no insights into the location of neurons responsible for rhythmogenesis. It is equally "conceivable" that the excited neurons provide a tonic drive to the neurons without being involved in the generation of active expiration. These tonic neurons could be located at a distance from the presumed rhythmogenic core.

We have included the possibility of tonic excitation in the technical considerations section:

“However, our study did not include recording from late-E neurons following bicuculline injections, preventing us from definitively confirming whether the observed activity stems from late-E neuronal excitation or the potentiation of a tonic drive, particularly in the rostral areas.”

(5) It is intriguing that some of their injections (Fig.2D) evoked postinspiratory activity. This interesting finding should be discussed as it could provide important insights into the coordination of the different phases of expiration.

Thanks for the suggestion. We have included the following to the discussion:

“Therefore, although it has previously been described, the exact mechanism by which this post-I ABD activity is generated is unclear. This late-E/post-I pattern of activity is similar to what has been observed in in vitro preparations and in vivo recordings in juvenile rats (Janczewski et al., 2002; Janczewski et al., 2006).

“Therefore, although it has previously been described, it is currently unknown the exact mechanism by which this post-I activity in the ABD muscles is generated. For example the interplay between the rostral pFL and brainstem structures generating post-inspiratory activity, such as the proposed post-inspiratory oscillator (PiCo; Anderson et al., 2016) or pontine respiratory networks, could be reasonably involved in this process.”

(6) The authors conducted bilateral disinhibition of the pFL, but only a unilateral photomicrograph was shown. Figure 2 should include a representative bilateral photomicrograph along with a scatter plot for clarity and completeness.

We have edited figure 2 to include representative images of bilateral injections.

(7) Regarding the Bicuculline injections in the Methods section: Aside from specifying exactly what type of bicuculline was used, the authors should provide more information about the pFL location and landmarks used, including the missing medial-lateral coordinate. The fluorobead spread of approximately ~300 µm, as observed in Figure 2C, is crucial for the interpretation of the results and should be detailed. An alternative approach could involve e.g. calculating the area covered by fluorobeads in each group.

We have included the following in the text:

“Each rat was injected at 2.8 mm lateral from the midline and at a specific RC coordinate based on the following groups: -0.2 mm from the caudal tip of the facial nucleus (VIIc) (n=5), +0.1 mm from VIIc (n=7), +0.4 mm from VIIc (n=5), +0.6 mm from VIIc (n=6), +0.8 mm from VIIc (n=5)”

“These findings strongly suggest that bicuculline specifically activated cells within the vicinity of the injection sites which spread ~300 ìm (Figure 2C, horizontal lines) and did not activate PHOX2B+ cells in the RTN area, beyond their baseline level of activity.”

(8) In the Experimental Protocol, the authors should provide more details on how the parameters were determined. For example, specify the number of cycles included for Dia frequency/amplitude, Abd frequency/amplitude, and with regards to the averaging process, the authors should specify over how many cycles they obtained an average for Dia/Abd activity time and AUC. The authors should also provide information on the number of bicuculline injections that they repeated to average these values and they should report the coefficient of variation for repeated injections. Please clarify the method used to calculate AUC, considering the non-linear nature of the activity.

Only one bicuculline injection per rat was performed and the number of rats used for each injection site is indicated in the methods as follows:

“Each rat was injected at 2.8 mm lateral from the midline and at a specific RC coordinate based on the following groups: -0.2 mm from the caudal tip of the facial nucleus (VIIc) (n=5), +0.1 mm from VIIc (n=7), +0.4 mm from VIIc (n=5), +0.6 mm from VIIc (n=6), +0.8 mm from VIIc (n=5), and CTRL (n=7). We recorded the physiological responses to the injection for 20-25 min.”

We have clarified in the methods section the following:

“Respiratory data was tracked in time bins of 2-minute duration from the baseline period prior to injections and spanned 20 min of recording post-injection. Mean-cycle measurements for each signal were computed by averaging values across all cycles within a given time bin.”

Additional clarifications have been added:

“We then used the average calculations of respiratory rate (RR), tidal volume (VT), Minute Ventilation (Ve), expiratory ABD amplitude, expiratory ABD area, VO2, VE/VO2 to obtain values relative to the baseline period. Peak responses were identified as the time bin that produced the strongest changes relative to baseline.”

“Mean-cycle measurements for each signal were computed by averaging across all cycles within a given time bin. (~300 cycles in baseline, ~100 cycles per response time bin). We then used the average calculations of respiratory rate (RR), tidal volume (VT), Minute Ventilation (Ve), expiratory ABD amplitude, expiratory ABD area, VO2, VE/VO2 to obtain values relative to the baseline period. Peak responses were identified as the time bin that produced the strongest changes relative to baseline.”

“The Area under the curve (AUC) was measured during baseline and was subtracted from the corresponding AUC of the response for each time bin (Figure 1C). This AUC measure was computed as the sum of the signal in a given respiratory phase as all signals were sampled at the same rate. Note that areas calculated below the zero- (0) line, as would be expected from a negative airflow during expiration, yields negative AUC values.”

(9) The authors should explain how oxygen consumption was calculated-did it involve the Depocas & Hart (1957) formula? Please provide information on expiratory CO2, whether ventilation was adjusted to achieve consistent CO2 levels across animals, and ideally specify the end-tidal CO2 range for the experiments. Discuss the rationale behind the chosen CO2 levels and whether CO2-dependent pFL activity could have influenced results.

We have clarified in the measurement in the methods as follows:

“The gas analyzer measured fractional concentration of O2. Based on this and the flow rate at the level of the trachea (minute ventilation), we calculated O2 consumption according to Depocas and Hart (1957).”

We have also added to the methods section:

“During the entire experimental procedure, rats breathed spontaneously and end tidal CO2 was not adjusted through the experimental protocol.”

In terms of the CO2-dependent pFL activity possibly influencing the results: by inducing active expiration in conditions in which there is no physiological demand for it (i.e. no hypoxia or hypercapnia), it is likely that pCO2 is reduced, overall decreasing the drive for ABD activity which would suggest that our results are likely an underestimation of the response that would have been produced if we maintained the CO2 levels constant.

(10) The authors should address the discrepancy in fos-activated neurons between the control (44 neurons) and experimental animals (90-120 neurons per hemisection). Please explain the activation in the control group. Please also provide insights into how the authors interpret this difference in cfos-activated neurons between control and experimental groups.

The following paragraph has been added to the discussion:

“The assessment of cellular activity, quantified through cFos staining, unveiled the existence of basal activity in control rats. This observed baseline activity is likely emanating from subthreshold physiological processes within the parafacial area which do not culminate in ABD activity. Analysis of the cFos staining confirmed focal activation of neurons in the pFL of rats injected with bicuculline and minimal cFos expression in the PHOX2B+ cells in all groups as compared to the control group. These results confirm the very limited mediolateral spread of the drug from the core site of injection and back previous findings supporting the hypothesis that the majority of PHOX2B+ cells are more ventrally located in the parafacial area (pFV, Huckstepp et al., 2015) and PHOX2B+ cell recruitment is not necessary for active expiration (de Britto & Moraes, 2017; Magalhães et al., 2021).”

(11) In Figure 8, the authors plotted the relationship of each cycle correlated to the normalized area. Have you also calculated the same late-E, inspiratory, and post-I to fR or VT separately?

No, we only did the separated breathing phase (late-E, I, Post-I) analysis in the calculations of the DIA, airflow and ABD area, as well as on the Euclidean and Mahalanobis distances.

Minor comments:Is there any specific reason for conducting these experiments exclusively in males?

No, we usually use male rats for this type of experiments. We use both male and female rats for other studies that concern the effects of sex hormones but in this case, we performed experiments only in male rats.

Page 13, Line 320: What is the duration of the bicuculline-induced effects?

This information is included in the results section as follows:

“Similarly, the ABD response duration was longer at the two most rostral locations (+0.6 mm = 17.6 ± 2.7 min; +0.8 = 17.1 ± 3.3 min) compared to the most caudal group (-0.2 mm = 2.4 ± 1.1 min; One-Way ANOVA p = 0.043; Tukey -0.2 mm vs +0.6 mm: p = 0.048; -0.2 mm vs +0.8 mm: p = 0.041; Figure 3E).”

Page 16, Line 400: Is there a rationale for the high tidal volume (VT) observed in these animals? A baseline VT of 7 ml/kg appears notably elevated.

Please note that rats were vagotomised and spontaneously breathing, hence the tidal volume is increased compared to non-vagotomised rats as seen in previous studies (Ouahchi et al., 2011).

Figure 2D: Could you provide longer recordings? Additionally, incorporating diaphragm (Dia) recordings would enhance the interpretation of abdominal (Abd) recordings.

Figure 3 A has a representative example of the 20 minute recordings for each location.

Page 18, Line 458: Please rectify "Dunn: p , 0.001" to the appropriate format, perhaps "Dunn: p < 0.001."

Thank you, edited.